# Formyl peptide receptor 2 determines sex-specific differences in the progression of nonalcoholic fatty liver disease and steatohepatitis

Chanbin Lee [1], Jieun Kim[1], Jinsol Han [1], Dayoung Oh[1], Minju Kim[1], Hayeong Jeong [1], Tae-Jin Kim[1,2], Sang-Woo Kim [1,2], Jeong Nam Kim [1,3], Young-Su Seo[1,3], Ayako Suzuki[4], Jae Ho Kim [5] & Youngmi Jung [1,2✉]

Nonalcoholic fatty liver disease (NAFLD) is an important health concern worldwide and progresses into nonalcoholic steatohepatitis (NASH). Although prevalence and severity of NAFLD/NASH are higher in men than premenopausal women, it remains unclear how sex affects NAFLD/NASH pathophysiology. Formyl peptide receptor 2 (FPR2) modulates inflammatory responses in several organs; however, its role in the liver is unknown. Here we show that FPR2 mediates sex-specific responses to diet-induced NAFLD/NASH. NASH-like liver injury was induced in both sexes during choline-deficient, L-amino acid-defined, high-fat diet (CDAHFD) feeding, but compared with females, male mice had more severe hepatic damage. Fpr2 was more highly expressed in hepatocytes and healthy livers from females than males, and FPR2 deletion exacerbated liver damage in CDAHFD-fed female mice. Estradiol induced Fpr2 expression, which protected hepatocytes and the liver from damage. In conclusion, our results demonstrate that FPR2 mediates sex-specific responses to diet-induced NAFLD/NASH, suggesting a novel therapeutic target for NAFLD/NASH.

[1] Department of Integrated Biological Science, College of Natural Science, Pusan National University, Pusan 46241, Republic of Korea. [2] Department of Biological Sciences, College of Natural Science, Pusan National University, Pusan 46241, Republic of Korea. [3] Department of Microbiology, College of Natural Science, Pusan National University, Pusan 46241, Republic of Korea. [4] Division of Gastroenterology and Hepatology, Duke University, Durham, NC, USA. [5] Department of Physiology, Pusan National University School of Medicine, Pusan National University, Yangsan 50612, Republic of Korea. ✉email: y.jung@pusan.ac.kr

Nonalcoholic fatty liver disease (NAFLD), one of the most common disorders in the world, and involves a spectrum of liver diseases that range from simple steatosis to its progressive form, nonalcoholic steatohepatitis (NASH)[1]. Approximately 25–30% of patients with NAFLD have NASH, which may lead to the development of cirrhosis and complications, such as hepatocellular carcinoma (HCC)[2]. To overcome NAFLD, many studies have been conducted and achieved tremendous research advancements in the NAFLD field. However, our understanding of NAFLD remains insufficient[3]. One of the major challenges in overcoming NAFLD is the heterogeneity in NAFLD risk profiles[4]. Accumulating evidence demonstrates that the prevalence and severity of NAFLD are affected by sex and age[5]. In adult populations, NAFLD prevalence is higher in men than in premenopausal women (or aged ≤ 50–60 years), while they tend to become more common in women after menopause (or aged ≥ 50–60 years)[6]. In South Korea, the prevalence of NAFLD was higher in men than women, but it increases in postmenopausal women compared with premenopausal women (41.1 vs 25.8/17%, respectively)[7]. In a cohort of Japanese subjects, the prevalence of NAFLD in men and post-menopausal women (24 and 15%, respectively) is higher than premenopausal women (6%)[8]. In South China, the prevalence of fatty liver disease, which includes 87% of NAFLD, is significantly lower in women than men under the age of 50 years (22.4 vs 7.1%, respectively), but women start to have higher prevalence rate compared to men over the age of 50 years (27.6 vs 20.6%)[9]. Furthermore, postmenopausal women on hormone replacement therapy (HRT) have a lower prevalence of NAFLD compared to postmenopausal women who are not on HRT, indicating that estrogen protects women from NAFLD development[10]. However, the mechanisms underlying sex differences and the action of estrogen in NAFLD remain largely unknown.

Formyl peptide receptor 2 (FPR2), which belongs to the G protein-coupled receptor family, is known as an important mediator in inflammatory and immune responses[11]. FPR2 is involved in multiple diseases, including bacterial infection, inflammation, asthma, Alzheimer's disease, and cancer[12]. Several studies have reported that FPR2 ligands, such as lipoxin A4 (LXA$_4$) and annexin A1 (AnXA1), have anti-inflammatory properties and mitigate hepatic damage by inhibiting hepatic inflammation in obese mice[13,14]. It has also been shown that FPR2-deficient mice have enhanced infiltration of immune cells with increased levels of interleukin (IL)-6, a proinflammatory cytokine, in liposaccharide (LPS)-stimulated liver injury[15]. These findings indicate that FPR2 is associated with hepatic inflammation. However, whether and how FPR2 is involved in NAFLD pathophysiology is poorly understood.

In the present study, we established a mouse model of NAFLD by adopting a choline-deficient, L-amino acid-defined, high-fat diet (CDAHFD) and examined sex differences in CDAHFD-fed mice. Given that inflammation is a common pathological feature of NAFLD and FPR2 modulates inflammatory responses, we investigated the role of FPR2 in NAFLD and the sex-specific progression of NAFLD. We demonstrate that Fpr2 is highly expressed in the livers of female mice and its expression regulated by estradiol plays a crucial role in protecting female mice from NAFLD/NASH development.

## Results

**CDAHFD-induced human-like NAFLD/NASH causes more hepatic damage in male mice than in female mice.** To mimic the pathogenesis of human NAFLD progression into NASH-HCC and investigate sex differences in NAFLD, we utilized a mouse model in which the mice of both sexes were fed a CDAHFD for 6,

12, and 36 weeks (Supplementary Fig. 1a). Gain of body weight (BW) tended to be lower in the CDAHFD-fed groups than the chow-fed groups (Supplementary Fig. 1b). Gross macroscopic images of the livers from chow-fed male and female mice showed healthy livers during feeding periods (Supplementary Fig. 1c). However, CDAHFD-fed male mice had enlarged and pale-colored livers compared with chow-fed mice, while the livers of CDAHFD-fed female mice exhibited a relatively smooth macroscopic appearance at 6 and 12 weeks. At 36 weeks of CDAHFD treatment, all male mice developed multiple nodules, whereas only two mice among five female mice developed multiple nodules (Supplementary Fig. 1d). Moreover, the nodule size was significantly larger in male mice than in female mice.

Lipid accumulation was observed in CDAHFD-fed mice at 6 weeks, and male mice had more and larger lipid droplets than female mice, as indicated by H&E staining (Fig. 1a). In CDAHFD-fed male mice, hepatic cell death was first observed at 12 weeks, and enhanced inflammation, hepatocyte ballooning, and nodule lesions displaying atypical adenomatous hyperplasia (marked by a circle) were detected at 36 weeks. CDAHFD-fed female mice exhibited relatively less hepatocellular death and smaller nodule lesions than CDAHFD-treated male mice. Abnormal histomorphological changes were rarely observed in the chow groups. The ratio of liver weight to body weight (LW/BW) and serum levels of alanine aminotransferase (ALT) and aspartate aminotransferase (AST) significantly increased during CDAHFD feeding, and among CDAHFD-fed mice, the LW/BW ratios and AST levels were significantly higher in male than in female mice at 36 weeks (Fig. 1b and Supplementary Fig. 1e). The level of hepatic triglycerides (TG) was gradually elevated in male mice during CDAHFD feeding and was significantly higher in male mice than in female mice at 36 weeks post CDAHFD treatment (Fig. 1c). Because massive hepatocyte death is a key process involved in the pathogenesis of NASH[16,17], we examined hepatocyte death in these experimental models. Immunohistochemical (IHC) analysis of active Caspase-3, the active form of a key executor of cellular apoptosis, visualized the cells expressing Caspase-3 in the livers of CDAHFD-fed mice, and these cells were more apparent in male mice than in female mice (Fig. 1d, e). Chow-fed WT mice rarely had these cells in their livers. Therefore, these data confirm that CDAHFD generates an experimental animal model of NAFLD progression similar to that of human NAFLD/NASH in our experimental conditions, and male mice were more susceptible to this injury than female mice.

**CDAHFD-fed male mice have excessive hepatic fibrosis and inflammation.** Because inflammation and fibrosis have been implicated in the pathogenesis of NAFLD[18], we evaluated hepatic fibrosis and inflammation in these mouse models. The fibrotic markers, alpha-smooth muscle actin (α-Sma) and collagen 1 alpha 1 (Col1α1) significantly increased in CDAHFD-fed WT mice compared with chow-fed mice, and the expression of these genes was remarkably upregulated in CDAHFD-fed male mice compared to CDAHFD-fed female mice at 12 and 36 weeks (Supplementary Fig. 2a). Western blot analysis confirmed the RNA data, as indicated by higher expression of α-Sma and Col1α1 in CDAHFD-male mice than in CDAHFD-female mice (Fig. 2a). The hydroxyproline assay, a biochemical assay to measure quantitative collagen accumulation in the liver, showed an increase in collagen fibrils in the CDAHFD groups compared with the chow groups at 12 and 36 weeks and its highest amount in male mice at 36 weeks among all CDAHFD-treated mice (Fig. 2b). More deposition of collagen fibrils was also observed in the livers of the CDAHFD groups (Fig. 2c, left panel) than in the livers of the chow groups (Supplementary Fig. 3, left panel), as assessed by Sirius red staining. In particular, fibrous septa were apparent in the livers of CDAHFD-treated male mice at

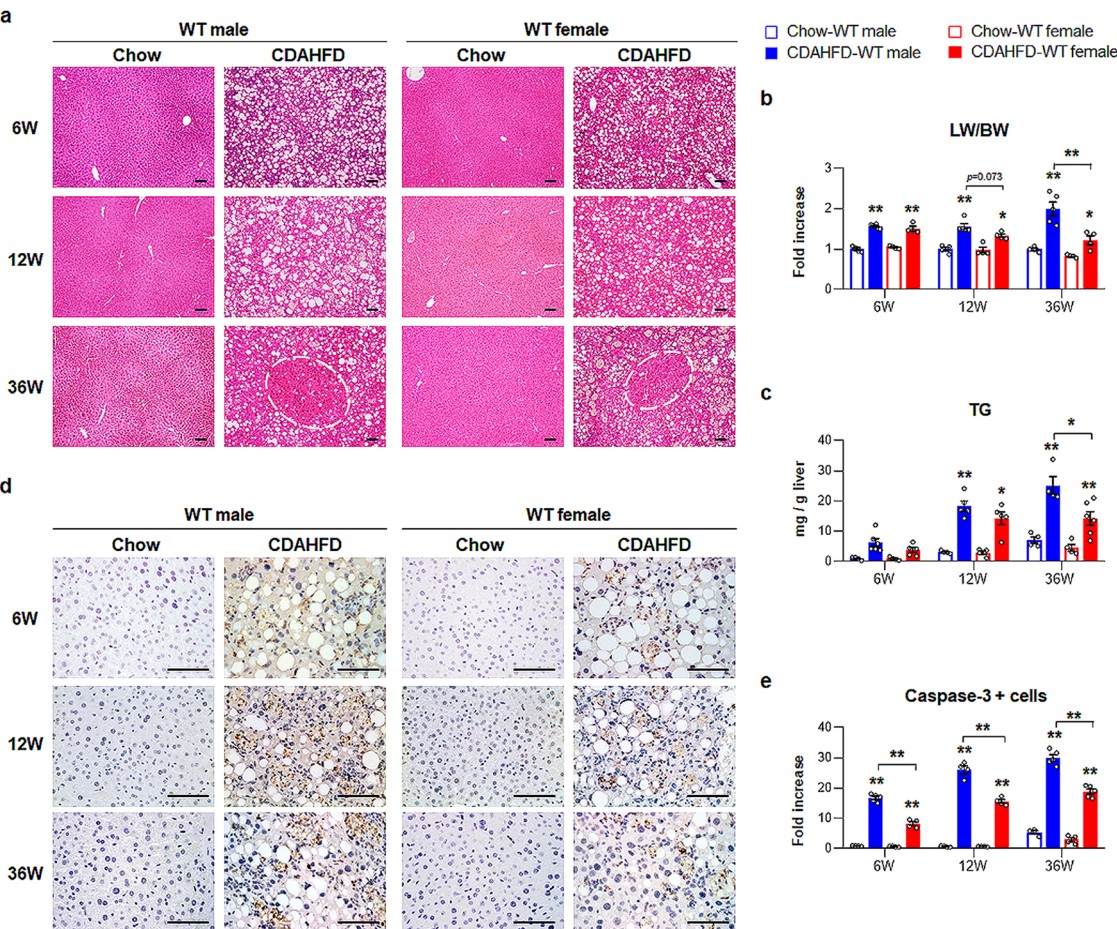

**Fig. 1 Hepatic injury in CDAHFD fed male and female mice. a** Representative images of the hematoxylin and eosin (H&E)-stained liver sections from wild-type (WT) male and female mice treated with chow or choline-deficient, L-amino acid-defined, high-fat diet (CDAHFD). White circles indicate the area of nodules on the liver. **b** The ratio of liver weight to body weight (LW/BW) and **c** hepatic triglycerides (TG) amount from these mice. Data represent the mean ± S.E.M. ($n \geq 4$/group, *$p < 0.05$, **$p < 0.005$ vs own control). **d** Representative images of active Caspase-3-immunohistochemical stained liver sections from these mice. All Scale bars show 50 µm. **e** Quantitative active Caspase-3 immunohistochemistry data from these mice. The numbers of Caspase-3-positive cells were counted per field and divided by the number of total hepatocytes per field. Data represent the mean ± S.E.M. ($n \geq 4$/group, *$p < 0.05$, **$p < 0.005$ vs own control). Gray circles represent individual data points. See Supplementary Data for statistical details.

3 weeks. Consistent with fibrotic changes in the CDAHFD groups, the expression of the proinflammatory markers tumor necrosis factor alpha (*Tnf-α*) and interleukin-6 (*Il-6*) increased in the CDAHFD group compared with the chow group, and the levels of these markers in CDAHFD-fed mice were significantly higher in male mice than in female mice at 36 weeks (Supplementary Fig. 2b). In addition, the serum levels of Tnf-α and Il-6 clearly showed significant upregulation in the CDAHFD groups and CDAHFD-male mice compared with the chow groups and CDAHFD-female mice, respectively (Fig. 2e). Immunostaining for F4/80 and CD68, markers of Kupffer cells, showed that these markers-positive cells accumulated in the livers of CDAHFD-fed mice, and these cells were more evident in male mice than in female mice (Fig. 2c, middle and right panel). Significant increase of these cells in male mice than in female was confirmed by quantitative analysis for F4/80- or CD68-positive cells (Fig. 2d). In the chow groups, these cells were rarely detected (Supplementary Fig. 3, middle and right panel). Therefore, these data revealed that female mice had less hepatic fibrosis and inflammation than male mice after CDAHFD injury.

**Increased expression of Fpr2 in female mice was alleviated during CDAHFD feeding.** Given that inflammation is closely related to NAFLD progression and FPR2 regulates inflammation[12,19],

we examined Fpr2 expression in these experimental mice. The livers of female mice had more baseline Fpr2 expression than those of male mice, and CDAHFD reduced Fpr2 expression in both sexes, as analyzed by qRT-PCR and western blot assays (Fig. 3a and Supplementary Fig. 4). IHC analysis confirmed that Fpr2-positive cells were more apparent in the livers of chow-fed female mice than in the livers of male mice, and the number of Fpr2-positive hepatocytic cells was significantly higher in the female mice than the male mice during chow feeding (Fig. 3b, c). CDAHFD reduced the number of Fpr2-expressing cells in both male and female, but Fpr2 expression was significantly upregulated in female mice compared with male mice. Interestingly, higher expression of Fpr2 in chow-fed female mice gradually decreased as the female mice aged (Fig. 3b, c, and Supplementary Fig. 4). Hence, serum estradiol levels were measured in these mice. Serum estradiol was consistently decreased in female mice with age, while the amount of estradiol in males remained constant (Fig. 3d). In addition, the estradiol level was lower in the CDAHFD-fed female mice than in chow-fed female mice at 6 weeks. Based on the decreasing patterns of both Fpr2 and estradiol in chow-fed female mice, we analyzed the relationship of Fpr2 with estradiol and found a positive correlation between hepatic expression of *Fpr2* and serum estradiol levels in these female mice (Fig. 3e). To determine if a similar correlation might be detected in humans, we analyzed hepatic *FPR2* expression in humans using published microarray data

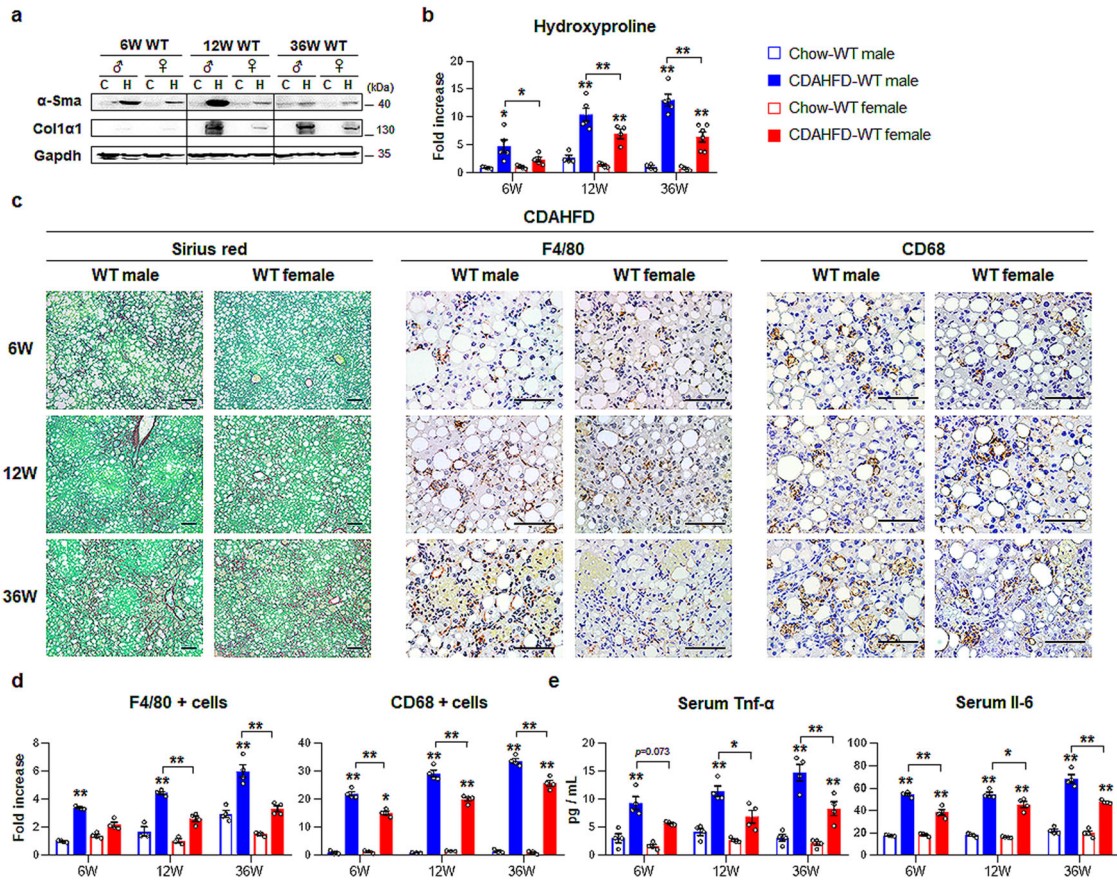

**Fig. 2 Enhanced liver fibrosis and inflammation in CDAHFD-fed male than female. a** Western blot analysis of hepatic alpha-smooth muscle actin (α-Sma) and collagen 1 alpha 1 (Col1α1) in wild-type (WT) male and female mice treated with chow or choline-deficient, L-amino acid-defined, high-fat diet (CDAHFD). Each lane contains protein lysates pooled from representative three mice per group with equal concentration. Glyceraldehyde 3-phosphate dehydrogenase (Gapdh) was used as internal control. The data shown represent one of three experiments with similar results. **b** Hepatic hydroxyproline content in liver tissues from representative mice per each group. Data represent the mean ± S.E.M. ($n \geq 4$/group, *$p < 0.05$, **$p < 0.005$ vs own control). **c** Representative images of Sirius red- (left panel) and F4/80- (middle panel) and CD68-stained (right panel) liver sections from the CDAHFD-WT groups (Scale bar, 50 μm). **d** Quantitative F4/80- or CD68-stained data from these WT mice. F4/80- or CD68-positive Kupffer cells were quantified by dividing the total numbers of positive cells by the total number of Kupffer cells. Data represent the mean ± S.E.M. ($n \geq 4$/group, *$p < 0.05$, **$p < 0.005$ vs own control). **e** Levels of serum Tnf-α and Il-6 in representative mice from each group. Data represent the mean ± S.E.M. ($n \geq 4$/group, *$p < 0.05$, **$p < 0.005$ vs own control). Gray circles represent individual data points. See Supplementary Data for statistical details.

(GEO access number: GSE66676)[20]. The age of human included in the cohort ranges from 13 to 19 and represents the group having an increasing level of estrogen in girls compared with boys. Hepatic *FPR2* expression was significantly higher in healthy females than healthy males (Supplementary Fig. 5), while its level was dramatically reduced in female patients with NAFLD and similar with it in male patients. There was no difference in *FPR2* expression between healthy men and male patients with NAFLD. In line with the human liver, these findings indicate that Fpr2 is highly expressed in the livers of healthy females and is downregulated with estrogen reduction and in the damaged liver, suggesting that distinct expression of Fpr2 in female mice might be associated with resistance to NAFLD-induced damage in women.

**Fpr2 deletion exacerbates liver damage in female mice.** To investigate whether Fpr2 was associated with the sex disparities in CDAHFD-induced NAFLD, Fpr2-deficient (KO) mice were treated with CDAHFD. After confirming the absence of Fpr2 expression in the livers of KO mice that were fed either chow or CDAHFD (Supplementary Fig. 6), the liver response to CDAHFD was investigated in these mice (Supplementary Fig. 7a).

Macroscopic examination showed that the livers of the chow-fed KO mice seemed to be normal, whereas the livers of the CDAHFD-treated KO mice began to exhibit a rough surface at 12 weeks and developed multiple nodules at 36 weeks in both sexes (Supplementary Fig. 7c). The number of mice with nodules was strikingly elevated in KO female mice compared with WT female mice after injury (Supplementary Fig. 1d, 7d). With an increase in nodule formation in CDAHFD-fed KO female mice, the size of the nodular lesions in these mice was larger than that of CDAHFD-fed WT female mice and similar to that of KO male mice exposed to CDAHFD. H&E staining showed that chow-fed KO mice had normal morphology, except for a slight infiltration of immune cells and accumulation of fatty hepatocytes in the livers of KO female mice, whereas CDAHFD-fed KO female and male mice at 36 weeks had severe histomorphological hepatic damage, such as excessive accumulation of macrovesicular lipid droplets, massive hepatocyte death, enhanced inflammation and atypical adenomatous hyperplasia, similar to that observed in WT male mice treated with CDAHFD at 36 weeks (Fig. 4a). Gain of BW tended to be lower in the CDAHFD-fed KO groups than the chow-fed KO groups (Supplementary Fig. 7b). LW/BW ratios,

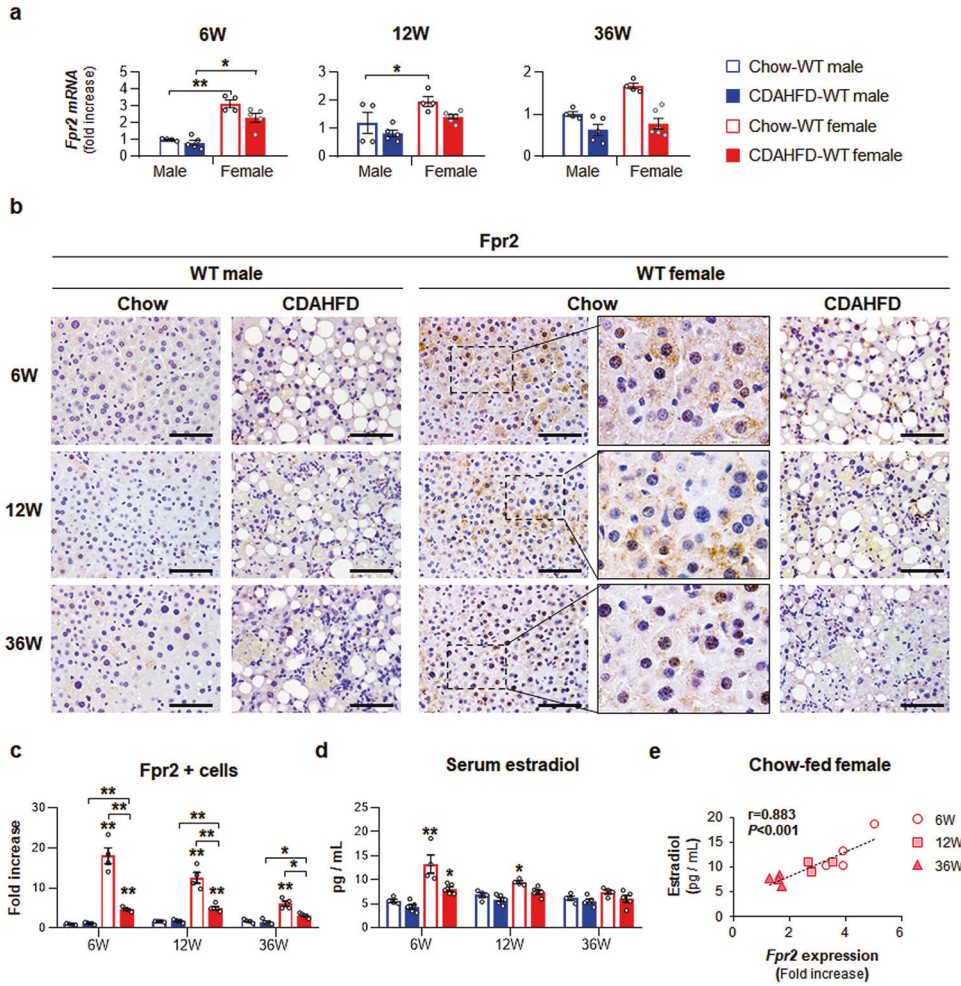

**Fig. 3 Distinct Fpr2 expression in livers of female mice. a** qRT-PCR for hepatic *formyl peptide receptor 2* (*Fpr2*) from WT male and female mice fed chow or CDAHFD. Data represent the mean ± S.E.M. ($n \geq 4$/group, *$p < 0.05$, **$p < 0.005$ vs own control). **b** Representative Fpr2-immunostained liver images from these mice. Magnified images of Chow-WT female in right panel are shown at X100 (Scale bar, 50 μm). **c** Quantitative Fpr2-stained data from these mice. The numbers of Fpr2-positive hepatocytic cells were counted per filed and divided by the number of total hepatocytes per field. Data represent the mean ± S.E.M. ($n \geq 4$/group, *$p < 0.05$, **$p < 0.005$ vs own control). **d** Amount of serum estradiol in these mice. Data represent the mean ± S.E.M. ($n \geq 4$/ group, *$p < 0.05$, **$p < 0.005$ vs own control). **e** Pearson's correlation between serum estradiol and Fpr2 expression of Chow-WT female mice ($n = 4$/ group, $r = 0.883$, $p < 0.001$). Gray circles represent individual data points. See Supplementary Data for statistical details.

serum levels of AST/AST, and hepatic TG levels were significantly elevated in the CDAHFD-KO groups (Fig. 4b, c and Supplementary Fig. 7e). In particular, in the CDAHFD-KO groups at 36 weeks, LW/BW ratios and hepatic TG levels were similar between male and female mice, and AST levels were significantly higher in female mice than in male mice. IHC analysis for Caspase-3 showed that Fpr2 defects increased baseline apoptosis in the livers of female mice and that CDAHFD promoted the accumulation of apoptotic cells in the livers of both KO male and female mice (Fig. 4d, e). These data demonstrated that Fpr2 deficiency caused more severe hepatic damage in CDAHFD-fed female mice, and hepatic injury was similar between KO male and KO female mice, unlike the hepatic response of WT mice exposed to CDAHFD.

**Increased fibrosis and inflammation in Fpr2-deficient female mice during liver damage.** Because Fpr2 deficiency worsened CDAHFD-induced liver injury in female mice as much as in male mice, we examined whether the lack of Fpr2 promoted hepatic fibrosis and inflammation in female mice. CDAHFD increased the RNA levels of the fibrotic markers *Col1α1* and *α-Sma* in KO male and female mice (Fig. 5a). Although the levels of these

factors were significantly enhanced in WT male mice compared with WT female mice at 12 and 36 weeks after CDAHFD (Supplementary Fig. 2a), in the CDAHFD-KO groups, the expression of fibrotic markers was similar between male and female mice. The protein levels of Col1α1 and α-Sma and hydroxyproline contents notably increased in KO male and female mice during liver injury, and especially hydroxyproline contents were significantly elevated in KO female mice compared with KO male mice at 36 weeks after CDAHFD (Fig. 5b, c). Sirius red staining also confirmed excessive collagen in the CDAHFD KO groups compared with the chow-fed KO groups (Fig. 5d). In the CDAHFD-KO groups at 36 weeks, fibrotic nodules were observed in both sexes, and collagen fibrils penetrating the parenchymal area were distinct in the livers of female mice. Consistent with fibrotic expression in the injured KO mice, the inflammatory markers, Tnf-α and Il-6, were upregulated in both KO mice fed CDAHFD compared with the chow-KO groups (Fig. 6a, b). Their RNA levels tended to be or were significantly higher in female mice than in KO male mice, and even Tnf-α in serum was significantly higher in KO female mice than in KO male mice at 36 weeks post CDAHFD feeding. In addition, F4/80- or CD68-positive Kupffer cells were aggregated to surround dying

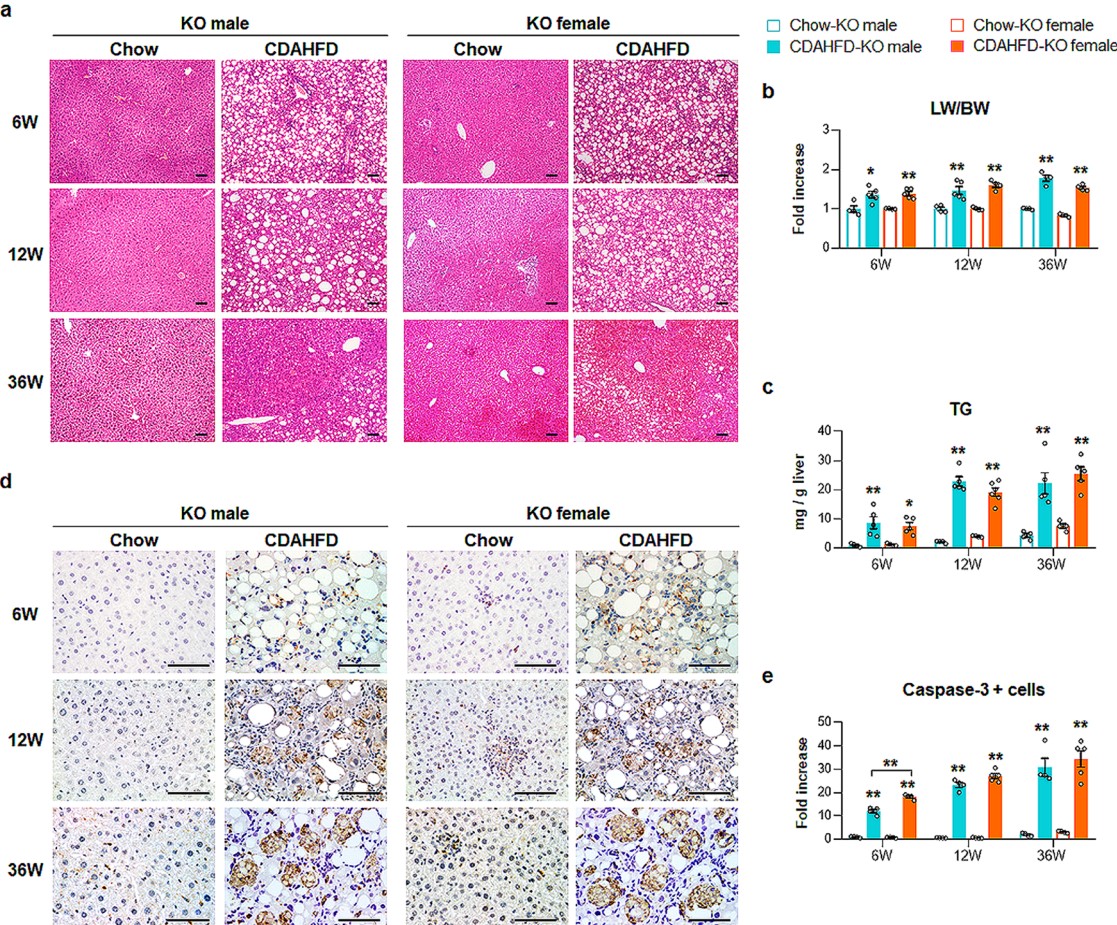

**Fig. 4 Fpr2-depleted female mice are vulnerable to liver damage. a** Representative images of H&E-stained liver sections from Fpr2-knockout (KO) male and female mice treated with chow or CDAHFD (Scale bar, 50 μm). **b** LW/BW and **c** hepatic TG level of these mice. Data represent the mean ± S.E.M. ($n \geq 4$/group, $*p < 0.05$, $**p < 0.005$ vs own control). **d** Immunohistochemical staining for active Caspase-3 in liver sections from representative mice from each group (Scale bar, 50 μm). **e** Quantitative active Caspase-3-stained data from these mice. Caspase-3-positive cells were quantified by counting the total number of Caspase-3-positive cells per field and dividing by the total number of hepatocytes per field. Data represent the mean ± S.E.M. ($n \geq 4$/group, $*p < 0.05$, $**p < 0.005$ vs own control). Gray circles represent individual data points. See Supplementary Data for statistical details.

hepatocytes in the CDAHFD-KO groups, and the number of F4/80- or CD68-positive cells were significantly higher in the CDAHFD groups than in the chow groups (Fig. 6c, d). Cells expressing F4/80 or CD68 were also observed in chow-fed KO female mice but not KO male mice (Supplementary Fig. 8). These findings suggest that Fpr2 deletion exacerbates liver fibrosis and inflammation in female mice during liver damage, suggesting that Fpr2 expression in the livers of female mice is associated with sex disparities in fibrosis and inflammation.

**Estradiol upregulates Fpr2 in hepatocytes and protects against lipotoxicity.** Accumulating evidence suggests that supplementation with estradiol has protective effects against NAFLD[21,22]. Given the protective effect of Fpr2 against liver injury in females and its positive correlation with estradiol, we investigated whether FPR2 was involved in the estrogen-mediated protective effect in hepatocytes. First, we identified what types of cells expressed FPR2 in the liver. Because hepatocyte-like cells expressed Fpr2 in the healthy livers of female mice (Fig. 3b), primary hepatocytes (pHEPs) were isolated from WT and KO mice. qRT-PCR and western blot results showed that hepatocytes from WT female mice had higher levels of Fpr2 than those from WT male mice (Fig. 7a, b). Double immunofluorescence staining for Fpr2 and Albumin, a marker of hepatocytes, also showed Fpr2 expression

in hepatocytes from WT females, as determined by confocal microscopy (Fig. 7c). Other types of cells from female mice, such as quiescent hepatic stellate cells (HSC), culture-activated HSC, and liver sinusoidal endothelial cells (LSECs), rarely had Fpr2 whereas Kupffer cells contained it (Supplementary Fig. 9a, b).

To determine whether Fpr2 expression protected hepatocytes from lipotoxicity, pHEPs isolated from female mice were exposed to 250 μM palmitic acid (PA), a lipotoxicity-inducing agent, for 24 h. As expected, PA led to a reduction in *Fpr2* expression, with decreased levels of glucose-6-phosphatase (*G6pc*), a marker of hepatocyte function, and decreased viability and an increased level of active Caspase-3 in pHEPs from WT female mice (Fig. 7d–f). Interestingly, Fpr2-deficient hepatocytes expressed lower level of *G6pc*, had lower cell viability and had higher expression of activated Caspase-3 at baseline than WT cells, and PA-induced damage exacerbated these injuries in Fpr2-depleted hepatocytes compared with Fpr2-expressing cells.

After confirming that hepatic expression of Fpr2 in female mice is involved in hepatocyte protection, we examined the association of Fpr2 with estrogen in these cells. Bioinformatics analysis using the TRANSFAC database predicted that estrogen response elements (EREs), which are recognized by estradiol-bound estrogen receptors (ERs), were present in upstream of the Fpr2 promoter. To determine whether Fpr2 had a binding site for ERs, a luciferase reporter assay was performed in WT or mutant-

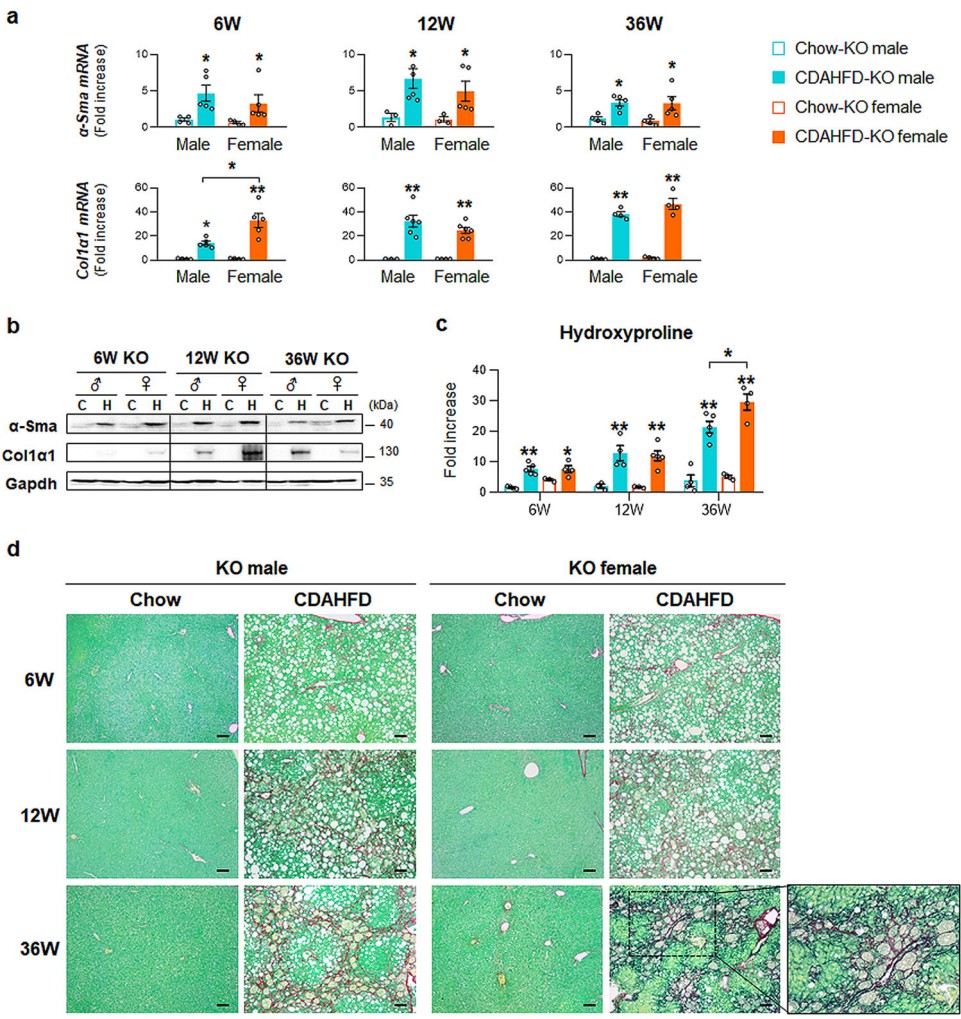

**Fig. 5 Fpr2 Deficiency leads to increased hepatic fibrosis in female mice fed CDAHFD. a** qRT-PCR analysis and **b** western blot analysis of hepatic α-Sma and Col1α1 from KO male and female mice treated with chow or CDAHFD for 6, 12, and 36 weeks. Data represent the mean ± S.E.M. ($n \geq 3$/group, $*p < 0.05$, $**p < 0.005$ vs own control). Each lane in immunoblots contains protein lysates pooled from representative three mice per group with equal concentration. Gapdh was used as internal control. The data shown represent one of three experiments with similar results. **c** Hepatic hydroxyproline content from these mice. **d** Representative images of Sirius red-liver sections from these mice. A magnified image of CDAHFD-KO female mice at 36 weeks is shown at X40 (Scale bar, 50 μm). Gray circles represent individual data points. See Supplementary Data for statistical details.

type (MT) ERE construct-transfected AML12 cells, a murine hepatocyte cell line; cells were treated with estradiol because estradiol treatment promotes nuclear localization of estrogen-bound ERs. The luciferase reporter assay showed that luciferase activity significantly increased in WT ERE construct-transfected cells but not MT construct-transfected cells after estradiol treatment (Fig. 8a). Estradiol also remarkably elevated luciferase activity in HepG2 cells transfected by pGL3 vectors containing ERE of the promoter of human FPR2, indicating that FPR2 had a binding site for ERs and that FPR2 expression is regulated by estradiol in both humans and mouse (Supplementary Fig. 10). In addition, estradiol treatment significantly elevated the expression of ER alpha (Erα), a major form of ERs in hepatocytes, in pHEPs from both WT and KO male mice compared with pHEPs treated with or without vehicle (Fig. 8b). Although the Fpr2 level was very low in WT male mice, Fpr2 expression was significantly upregulated in estradiol-treated pHEPs from these mice compared with vehicle-exposed cells. In pHEPs from KO male mice, Fpr2 was rarely detected, even though estradiol was administered to these mice. Hence, these data clearly demonstrate that Fpr2 expression in hepatocytes is stimulated by estradiol.

We next investigated whether estradiol-induced Fpr2 protected male pHEPs from damage (Fig. 8c). Compared with PA-treated pHEPs from WT male mice without estradiol, estradiol-treated cells showed significant upregulation of Fpr2, G6pc expression and cell viability and mitigation of cleaved Caspase-3 levels and intracellular ROS production, although these cells were exposed to PA. (Fig. 8d–g). Compared with vehicle-treated pHEPs from KO male mice, PA-treated cells from these mice had lower levels of G6pc and cell viability and higher amounts of active Caspase-3 and intracellular ROS, regardless of estradiol treatment. Taken together, these findings suggest that estrogen promotes Fpr2 expression, which protects hepatocytes from PA-induced injury.

**Fpr2 is involved in hepatic VLDL secretion and alleviates lipid accumulation.** Choline deficiency impairs very-low density lipoprotein (VLDL) secretion, and leads to fat accumulation[23,24]. However, female human and mice have a capacity for de novo biosynthesis of choline by phosphatidylethanolamine N-methyltransferase (PEMT), which makes them be less sensitive to choline deficiency[25,26]. PEMT is known to be regulated by estradiol[27]. Based on these findings, we examined whether FPR2

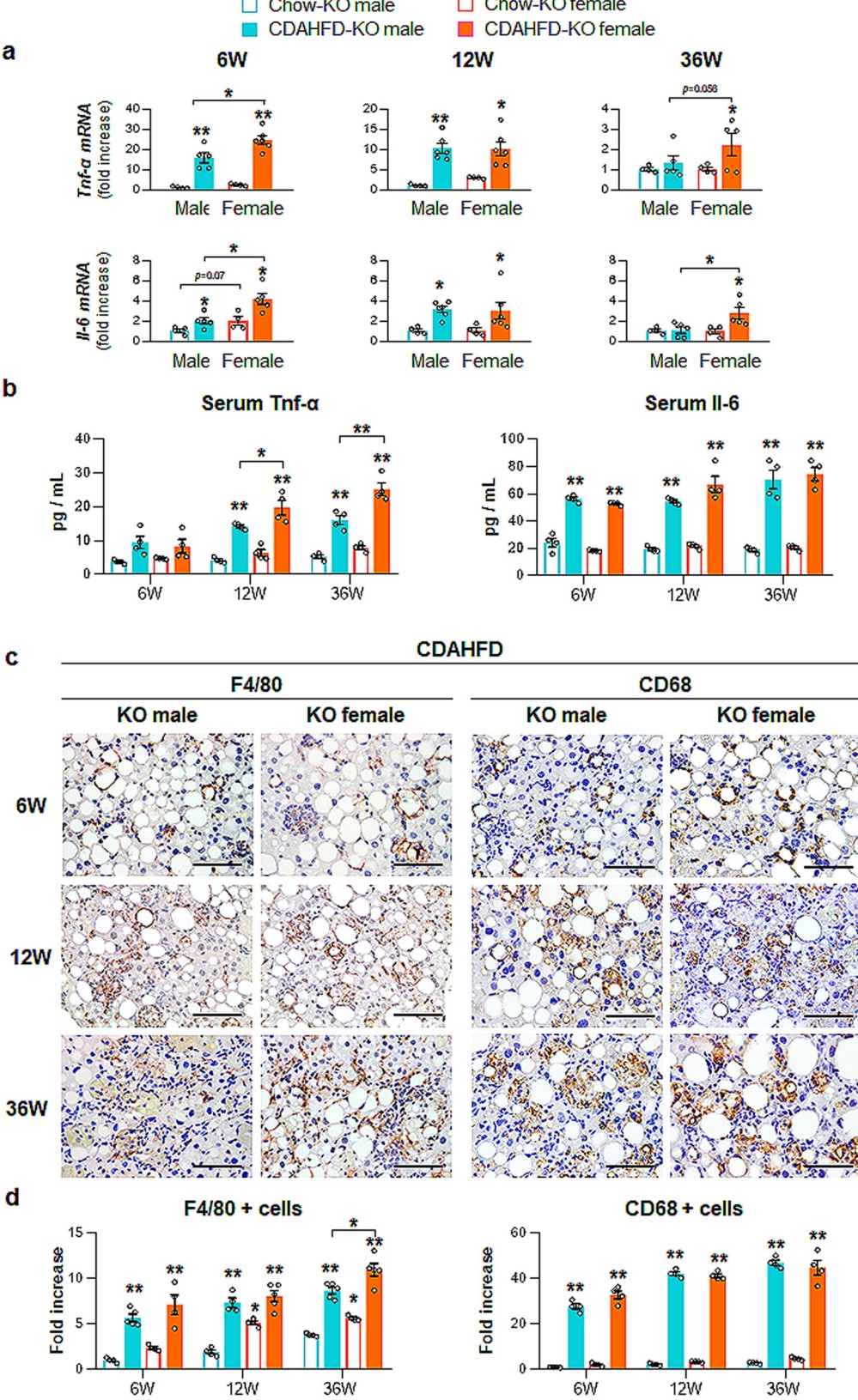

**Fig. 6 Lack of Fpr2 elevates inflammation in female mice during CDAHFD feeding. a** qRT-PCR analysis of inflammatory markers including *Tumor necrosis factor-alpha (Tnf-α)*, *Interleukin (Il-6)* in KO male and female mice treated with chow or CDAHFD for 6, 12, and 36 weeks. **b** Levels of serum Tnf-α and Il-6 from these mice. All data represent the mean ± S.E.M. ($n \geq 4$/group, *$p < 0.05$, **$p < 0.005$ vs own control). **c** Representative images of F4/80- or CD68- stained liver sections from these mice (Scale bar, 50 μm). **d** Quantitative F4/80- or CD68-stained data from these mice. F4/80- or CD68- positive Kupffer cells were quantified by dividing the total numbers of positive cells by the total number of Kupffer cells. Data represent the mean ± S.E.M. ($n \geq 4$/group, *$p < 0.05$, **$p < 0.005$ vs own control). Gray circles represent individual data points. See Supplementary Data for statistical details.

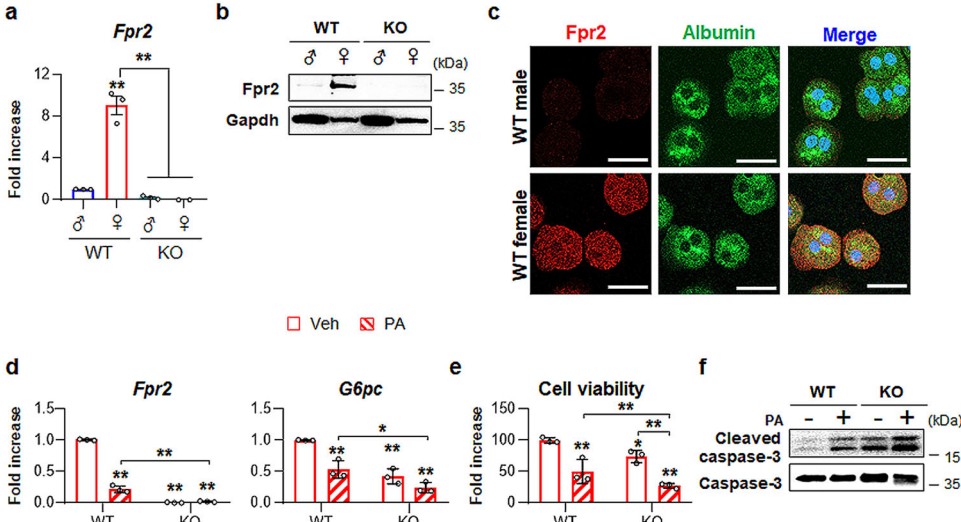

**Fig. 7 Higher expression of Fpr2 in the livers of female mice is related with hepatocyte protection. a** qRT-PCP analysis for *Fpr2* expression in primary hepatocytes (pHEPs) from WT (male *n* = 2, female *n* = 2) and KO mice (male *n* = 2, female *n* = 2). Total eight mice were employed in each hepatocyte isolation, and the experiments were replicated at least three times and the mean ± S.E.M. results are graphed (*$p < 0.05$, **$p < 0.005$ vs WT male-pHEPs). **b** Western blot analysis and **c** double immunofluorescent images of Fpr2 (red) with albumin (green) in these cells. Gapdh was used as internal control. DAPI (blue) was used as nuclear counterstaining. Data shown represent one of three experiments with similar results (Scale bar, 20 μm). **d** qRT-PCR analysis for *Fpr2* and *glucose-6-phosphatase* (*G6pc*) in, **e** cell viability of, and **f** western blot analysis of cleaved Caspase-3 and pro Caspase-3 in WT and KO female mice-isolated pHEPs treated with vehicle (Veh) or 250 μM of palmitate (PA). The data shown represent one of three experiments with similar results. The mean ± S.E.M. results obtained from three repetitive experiments are graphed (*$p < 0.05$, **$p < 0.005$ vs WT-Veh). Gray circles represent individual data points. See Supplementary Data for statistical details.

influenced VLDL secretion through PEMT, and reduced fat accumulation in hepatocytes, protecting them from PA injury. Estradiol-treated pHEPs from WT male mice containing higher *Fpr2* level showed significantly increased expressions of *Pemt* and VLDL secretion-related genes, apolipoprotein B (*ApoB*) and apolipoprotein C3 (*ApoC3*), compared with the vehicle-treated cells (Supplementary Fig. 11a). PA exposure downregulated *Pemt*, *ApoB*, and *ApoC3* in pHEP from WT males, but their expressions were significantly higher in the estradiol-treated pHEPs than the vehicle-treated cells. In pHEPs from KO mice, expressions of these genes were rarely induced by estradiol. In line with RNA data, Oil Red O staining showed that lipid droplets greatly accumulated in the PA-treated pHEPs from WT and KO male, whereas estradiol apparently lowered lipid droplets in PA-treated cells from WT mice, not from KO mice (Supplementary Fig. 11b). Given that estradiol hardly changed expressions of *Pemt* in pHEPs from KO mice, the data suggested that FPR2 might be involved in PEMT expression.

Expressional changes of *Pemt* and VLDL secretion-related markers were also examined in the NAFLD-like animal models. As expected, the expressions of *Pemt*, *ApoB*, and *ApoC3* significantly increased in the chow-fed WT female mice compared with the chow-treated WT male (Supplementary Fig. 12a). CDAHFD reduced their levels in both sexes, but their expressions were significantly or tended to be elevated in the CDAHFD-fed females than CDAHFD-treated males. However, the RNA levels of these genes were similar in in the KO female and KO male mice, regardless of diets (Supplementary Fig. 12b). KO female mice had a deficient Fpr2, not estradiol, and no expressional change of *Pemt* regulated by estradiol in these mice confirmed the in vitro data by indicating that Fpr2 impacted Pemt expression. Taken together, these results suggest that FPR2 is involved in VLDL secretion by affecting PEMT, and ameliorates liver damage.

**Fpr2-mediated estrogen protects the liver from CDAHFD-induced NASH**. To assess whether the protective effect of estrogen-mediated Fpr2 in hepatocytes occurs in vivo, WT male mice were supplemented with either estradiol (E2 group) or placebo pellets (P group) and were fed a chow diet or CDAHFD for 12 weeks (Supplementary Fig. 13a). Exogenous estradiol led to a significant increase in serum estradiol and upregulation of Fpr2 expression in WT male mice fed either a chow diet or CDAHFD compared with the control group (Fig. 9a–c and Supplementary Fig. 13f). As expected, estradiol treatment attenuated CDAHFD-induced liver damage, as evidenced by macroscopic examination of the liver, reduced LW/BW ratios and ALT/AST levels, and decreased deposition of fatty hepatocytes and active Caspase-3-positive cells (Fig. 9d and Supplementary Fig. 13b–d). In addition, the degree of hepatic fibrosis and inflammation were remarkably decreased in the CDAHFD-E2 group compared with the CDAHFD-P group, as assessed by Sirius red staining, hydroxyproline levels, quantification analyses of the fibrotic and inflammatory markers α-Sma, Col1α1, Tnf-α and Il-6, and IHC analysis of F4/80 (Fig. 9d–f and Supplementary Fig. 13e–g). In the chow group, estradiol hardly affected the liver compared with that of placebo-treated mice, although estradiol supplementation increased the serum level of estradiol in these mice (Fig. 9 and Supplementary Fig. 13).

To confirm whether estrogen-mediated Fpr2 expression reduced susceptibility to NASH progression in female mice, both ovaries were removed from WT female mice, and chow or CDAHFD was given to these female mice for 12 weeks (Supplementary Fig. 14a). The levels of both serum estradiol and hepatic Fpr2 were significantly reduced in ovariectomized mice (OVX) compared with sham-operated mice (Sham) (Fig. 10a–c and Supplementary Fig. 14f). In CDAHFD-treated female mice, liver injury and the accumulation of fatty hepatocytes and active Caspase-3-expressing cells were more

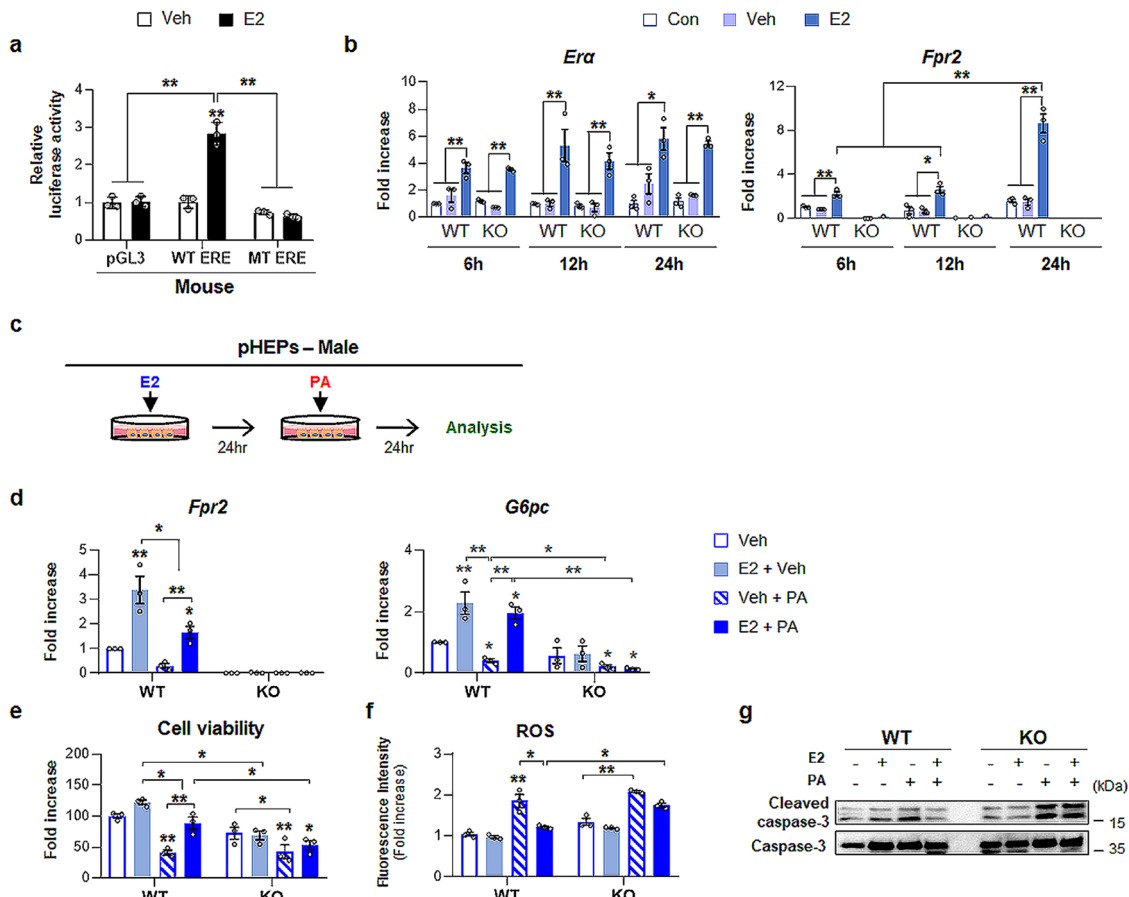

**Fig. 8 Estradiol protects hepatocytes from lipotoxicity by upregulating Fpr2. a** Dual luciferase reporter assay in AML12 cells transfected with a pGL3 vector having either WT- or mutant (MT-) ERE on the promoter region of Fpr2. These cells were exposed to either 100 nM estradiol (E2) or vehicle (Veh). Mean ± S.E.M. results are graphed (*$p < 0.05$, **$p < 0.005$ vs Veh). **b** qRT-PCR analysis for *Estrogen receptor alpha (Erα)* and *Fpr2* in primary hepatocytes isolated from WT and KO male mice treated with none (Con), vehicle (veh) or E2. The mean ± S.E.M. results obtained from three repetitive experiments are graphed (*$p < 0.05$, **$p < 0.005$ vs Con). **c** A scheme for cell experiment in which pHEPs treated with either vehicle or 100 nM of E2 were exposed to 250 μM of PA for 24 h. **d** qRT-PCR analysis for *Fpr2* and *G6pc*, **e** cell viability, **f** intracellular reactive oxygen species (ROS) levels in these cells. and **g** western blot analysis for cleaved Caspase-3 and pro-Caspase-3 The mean ± S.E.M. results obtained from three repetitive experiments are graphed (*$p < 0.05$, **$p < 0.005$ vs WT-Veh). The data shown represent one of three experiments with similar results. Gray circles represent individual data points. See Supplementary Data for statistical details.

evident in the OVX group than in the Sham group (Fig. 10d and Supplementary Fig. 14b–d). The degree of fibrosis and inflammation was notably enhanced in the CDAHFD-OVX group compared with the CDAHFD-Sham group (Fig. 10d–f and Supplementary Fig. 14e–g). In the chow groups, no significant changes between the OVX and Sham groups were observed, except for higher levels of serum Il-6 in the OVX group than in the Sham group (Fig. 10 and Supplementary Fig. 14).

To double-check the action of estrogen-regulated Fpr2 expression in NAFLD, ovariectomized WT female mice were supplemented estradiol (OVX-E2) or placebo pellets (OVX-P), then were fed chow or CDAHFD for 12 weeks (Supplementary Fig. 15a). Reduced levels of serum estradiol and hepatic Fpr2 expression significantly increased in the OVX-E2 compared with the OVX-P groups (Supplementary Figs. 15f and 16a–c). Compared with the CDAHFD-fed OVX-P group, the CDAHFD-treated OVX-E2 group had reduced liver damage and accumulation of Caspase 3-positive cells (Supplementary Fig. 15b–d and 16d). Estradiol supplementation also mitigated the enhanced hepatic fibrosis and inflammation in CDAHFD-given female mice with OVX (Supplementary Figs. 15e–g and 16d–f). In the chow groups, hepatic injuries by estradiol supplement were absent (Supplementary Figs. 15, 16). Taken together, these

findings suggest that FPR2 is involved in the hepatoprotective effects of estrogen against NAFLD progression.

## Discussion

The etiology of NAFLD is complex and influenced by multiple factors, such as obesity, insulin resistance, hyperlipidemia, and sex[1]. During recent decades, studies have shown sex disparities in NAFLD/NASH development, suggesting the importance of research on sex-specific responses to nonalcoholic liver damage[28,29]. Several epidemiological and experimental studies have shown that NAFLD is worse in men than in women and that NAFLD prevalence is higher in postmenopausal women than in premenopausal women, indicating that sex disparities in NAFLD are related to sex hormones, especially estrogen[30,31]. Consistent with these findings, our results showed that female mice were more resistant to NAFLD development than male mice during CDAHFD feeding for 36 weeks (Figs. 1, 2 and Supplementary Figs.1–3). The CDAHFD model was developed by Matsumoto's group[32,33]. They showed that CDAHFD feeding for 12–60 weeks induced progression from steatosis to NASH with fibrosis and subsequently led to tumorigenesis in mice, suggesting that this model reflects human NAFLD/NASH better than other models.

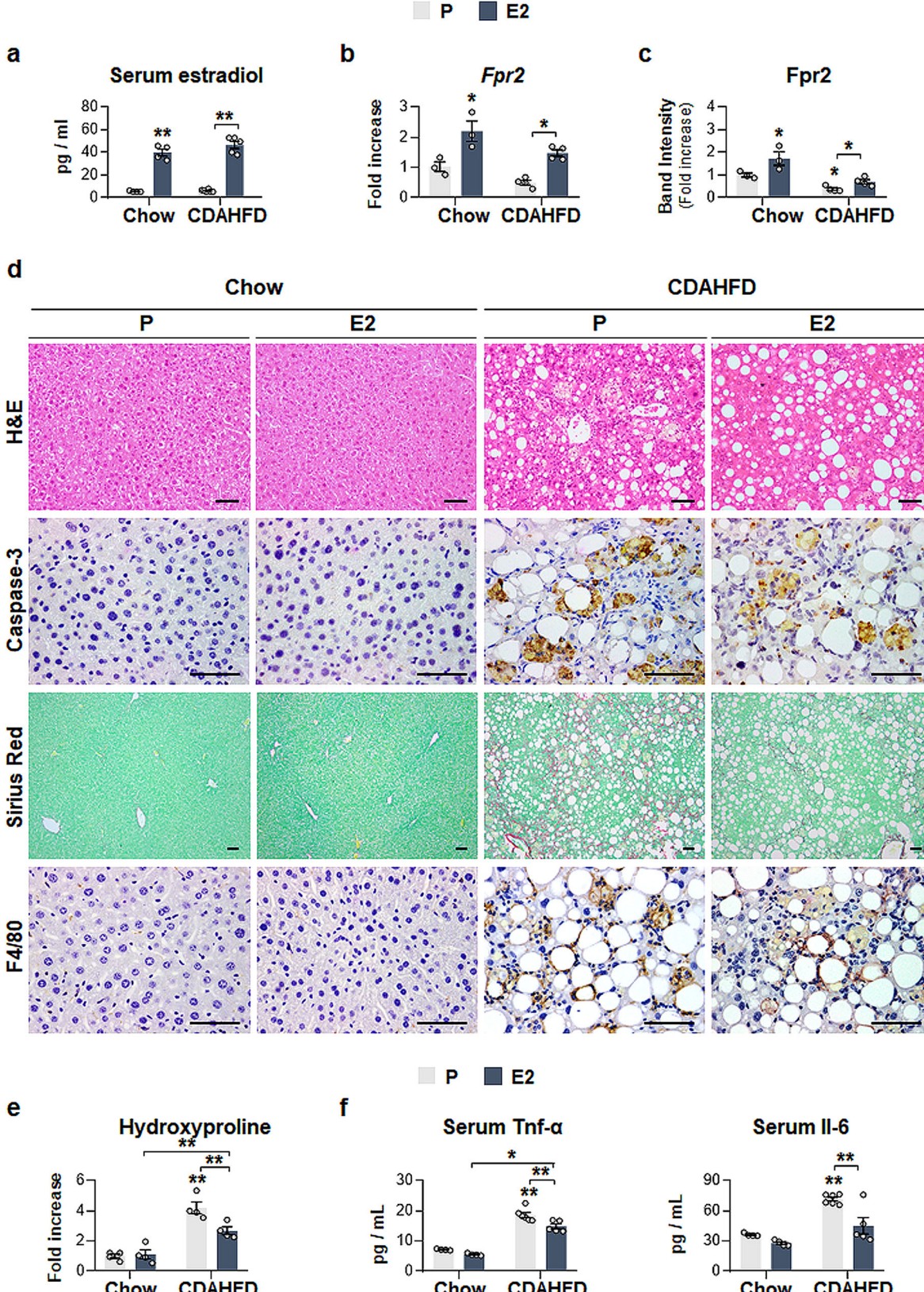

**Fig. 9 Estradiol implantation elevates Fpr2 expression, and alleviate liver damage in male mice. a** Amount of serum estradiol in, **b** qRT-PCR and **c** cumulative densitometric analysis for hepatic Fpr2 in placebo (P) or estradiol (E2)-supplemented male mice fed either chow or CDAHFD for 12 weeks. Band densities were normalized to the expression level of Gapdh, an internal control. Data represent the mean ± S.E.M. ($n \geq 3$/group, $*p < 0.05$, $**p < 0.005$ vs Chow-P). **d** Representative images of H&E-, Caspase-3-, Sirius red-, and F4/80-stained liver sections from these mice (Scale bar: 50 μm). **e** Hepatic hydroxyproline content and **f** serum Tnf-α and Il-6 level in these mice. Data represent the mean ± S.E.M. ($n \geq 3$/group, $*p < 0.05$, $**p < 0.005$ vs Chow-P). Gray circles represent individual data points. See Supplementary Data for statistical details.

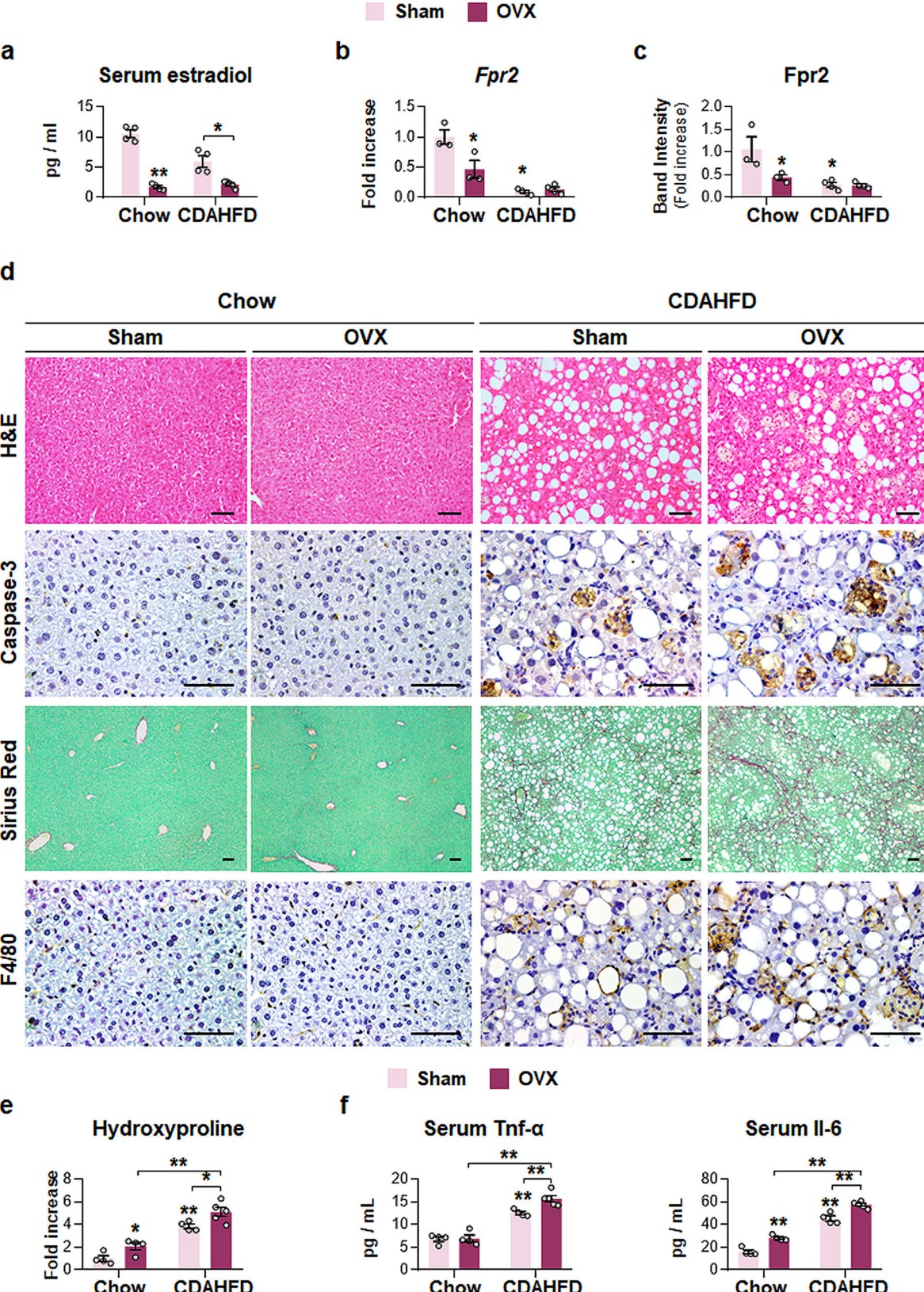

**Fig. 10 Ovariectomy decrease hepatic Fpr2 level, and increases liver injury in female mice fed CDAHFD. a** Level of serum estradiol in, **b** qRT-PCR and **c** cumulative densitometric analysis for hepatic Fpr2 in sham (Sham) or ovariectomized (OVX) female mice treated with either chow or CDAHFD for 12 weeks. Band densities were normalized to the expression level of Gapdh, an internal control. Data represent the mean ± S.E.M. ($n \geq 3$/group, *$p < 0.05$, **$p < 0.005$ vs Chow-Sham). **d** Representative images of H&E-, Caspase-3-, Sirius red-, and F4/80-stained liver sections from these mice (Scale bar: 50 μm). **e** Hepatic hydroxyproline content and **f** serum Tnf-α and Il-6 level in these mice. Data represent the mean ± S.E.M. ($n \geq 3$/group, *$p < 0.05$, **$p < 0.005$ vs Chow-Sham). Gray circles represent individual data points. See Supplementary Data for statistical details.

In the present study, we used this model to generate human-like NAFLD that exhibited macrovesicular steatosis, lobular inflammation, hepatocyte ballooning, and fibrosis and determined whether this model could display the sex differences observed in human NAFLD. CDAHFD-fed male and female mice showed comparable hepatic damage by 6 weeks, and their differences become clear after 12–36 weeks. CDAHFD-fed males had more severe hepatic damage, such as higher TG accumulation, hepatocyte death, inflammation, fibrosis, and even tumorigenesis, than CDAHFD-fed females (Figs. 1, 2 and Supplementary Figs. 1–3), indicating that our experimental NAFLD model reflects the NAFLD-NASH-HCC progression observed in humans.

FPR2, which modulates the inflammatory response, is reported to exert an anti-inflammatory effect in the liver[15]; however, the mechanism underlying the effect of FPR2 in the liver has not been fully elucidated. Most results have been obtained from in vitro studies and male-based in vivo studies. Therefore, examination of sex-specific FPR2 expression in the liver, beginning with the study of hepatic FPR2, has been overlooked. In the present study, we found a sex difference in hepatic Fpr2 expression. The basal level of hepatic Fpr2 was markedly higher in females than in males, and Fpr2 was mainly expressed in hepatocytes in the healthy livers of female mice (Figs. 3, 7a–c, and Supplementary Fig. 4). In human liver, hepatic FPR2 expression was also significantly higher in healthy female than in healthy males (Supplementary Fig. 5). In addition, our results demonstrated that hepatic Fpr2 was related to the protection of female livers against NAFLD progression. Female WT mice showed less susceptibility to CDAHFD than male mice, whereas female KO mice were vulnerable to liver injury, similar to male mice, which express lower levels of Fpr2 than females (Figs. 1–6 and Supplementary Figs. 1–9). Moreover, induced Fpr2 expression by exogenous estradiol protected male hepatocytes from lipotoxicity (Fig. 8d–g). Hence, it is possible that FPR2-mediated hepatocyte survival leads to reduced inflammation and fibrosis, which are commonly observed in NASH accompanied by massive hepatocyte death. In addition, we measured the level of LXA$_4$, one of Fpr2 ligands, in our experimental animal models. LXA$_4$ has been shown to have anti-inflammatory effects in the various disease models including acute liver failure and obesity[13,34,35]. Hepatic LXA$_4$ amount rarely had a significant difference between male and female mice (Supplementary Fig. 17a). However, LXA$_4$ was downregulated significantly in CDAHFD-fed male and female mice compared with chow-fed mice. Decrease of LXA$_4$ expression in the CDAHFD groups compared to the chow groups was also found in the experimental models in which Fpr2 was knockout or the estrogen level was altered artificially (Supplementary Fig. 17b–e). The data indicated that both male and female mice had LXA$_4$, but female mice with a higher level of Fpr2, LXA$_4$ receptor, could effectively reduce the liver damage.

Estradiol is the major circulating form of estrogen, which is produced by the ovaries and influences hormone-sensitive organs, including the liver[36]. Most estrogen functions are mediated by its two nuclear receptors, ERα and ERβ. Estrogens bind and activate ERs, and the estrogen/ER complex translocates into the nucleus, where it binds to specific DNA sequences known as EREs to regulate the transcription of target genes[37]. In the liver, ERα and ERβ are primarily expressed in hepatocytes and activated HSCs, respectively[38,39]. In the current study, we revealed that FPR2 is a downstream target of estrogen and that its expression is stimulated by estradiol (Figs. 8a–c, 9a–c, and 10a–c and Supplementary Fig. 10). In the female mice, the serum estradiol level progressively decreased as the expression of hepatic Fpr2 gradually decreased with age and was similar in 11-month-old male and female mice (Fig. 3e). Mice are sexually mature by 3–6 months of age and approach the endocrine equivalent to the human menopause transition by 9 months, and 10–14-month-old mice correspond to the middle age of humans (38–47 years), which is the perimenopause period at which estradiol levels gradually decrease[40]. Therefore, the parallel decreased estradiol level with the reduction in FPR2 seems to explain the increased incidence of NAFLD in postmenopausal women. Fpr2 induction in male mice by estradiol supplementation protected the liver, while Fpr2 suppression in female mice by ovariectomy-mediated estrogen depletion exacerbated liver damage during CDAHFD feeding. Estrogen implantation in female mice with OVX also restored hepatic Fpr2 level and alleviated CDAHFD-induced liver injury (Supplementary Figs. 15, 16). Therefore, these results indicated that estrogen modulates the hepatic level of FPR2, which regulates the liver response to damage.

Although Fpr2 in the RNA level was not significantly different between the chow- and CDAHFD-fed female mice (Fig. 3a), the amount of Fpr2 protein was significantly decreased (Fig. 3b, c and Supplementary Fig. 4). The different expressional patterns of Fpr2 in RNA and protein levels indicate that CDAHFD might influence Fpr2 expression at a post-transcriptional level. Pierdomenico et al.[41] have demonstrated that miR-181b directly binds to 3'-untranslated region of FPR2, and impacts FPR2 expression. In addition, it has shown that upregulated miR-181b lowers FPR2 expression by targeting FPR2, and reduced anti-inflammatory response caused by FPR2 suppression contributes to lung cystic fibrosis[42]. Furthermore, miR-181b expression in serum was higher in the patient with NAFLD than healthy controls[43]. Hepatic miR-181b was shown to be elevated in high-fat diet-fed mice compared with normal diet-fed mice, and inhibition of miR-181b suppressed accumulation of triglycerides[43]. In choline-deficient and amino acid-defined (CDAA) diet model, miR-181b was upregulated in mice fed with CDAA[44]. Given that miR-181b targeting FPR2 is upregulated in the NAFLD-like experimental animal models and patients with NAFLD, and Fpr2 protein is lower in the CDAHFD-fed female than the chow-fed female mice, it is possible that CDAHFD decreases Fpr2 expression at the post-transcriptional level by increasing miR-181b. However, the Fpr2 amount was still elevated in females compared with males during CDAHFD feeding (Fig. 3b, c), indicating that estrogen-stimulated Fpr2 seems to at least partially compensate for Fpr2 loss by CDAHFD, and prevents female mice from NAFLD progression. Further studies are required to verify the effect of miR-181b on FPR2 protein expression in NAFLD.

Choline is an essential nutrient in producing phosphatidylcholine (PC) required for VLDL synthesis[45]. Deficiency of choline results in excessive lipid storage in the liver by impaired lipid outflow from the liver[26,46]. However, it has been reported that human and rodent female have a lower risk for hepatic steatosis than male, because of higher expression of hepatic PEMT in females. PEMT is a transferase enzyme, which makes PC from phosphatidylethanolamine in the liver. PEMT expression is directly regulated by estrogen, and lipid accumulation is less in female than male because of PEMT[27]. In the CDAHFD-fed WT mice, Pemt was significantly upregulated in WT female mice compared with WT male mice, and increased the expression of VLDL secretion markers, possibly reducing the levels of TG and hepatic fat (Supplementary Fig. 12a). However, sex difference of hepatic Pemt expression was disappeared in KO mice, and CDAHFD-fed KO female mice had similar degrees of lipid accumulation with CDAHFD-treated KO male mice did (Fig. 4a, c and Supplementary Fig. 12b). In addition, estradiol exposure induced Pemt expression and reduced accumulation of lipid droplets in pHEPs from WT male mice, not the cells from Fpr2-KO male mice (Supplementary Fig. 11). Based on these findings, it is possible that FPR2 is involved in PEMT regulation by estradiol, and improves VLDL secretion through PEMT, leading to the

decreased fat accumulation in the liver. Influx of excessive free fatty acid (FFA) brings to oxidative stress and mitochondria dysfunction, resulting in massive hepatocytes death. And FFA level is known to be positively correlated with NAFLD severity[16,47,48]. Dying hepatocytes released several cytokines, such as PDGF, CTGF, TGF-β, and hedgehog, which promote inflammation and fibrosis in the liver[18,49]. Thus, reducing hepatic FFA levels is critical to treat NAFLD. Therefore, it is possible that the promoting effect of FPR2 on VLDL secretion through PMET leads hepatocytes to be resistant to lipotoxicity, contributing to the hepatocyte survival, subsequently reduction of inflammation and fibrosis. However, further studies are required to investigate the detailed interaction among FPR2, PEMT, and estrogen in modulating lipid metabolism in the liver.

In conclusion, our results demonstrate that FPR2 has anti-inflammatory roles in the liver and that its sex-specific expression is closely related to sex disparities in NAFLD/NASH development. Higher expression of FPR2 in female mice than in male mice makes females more resistant to NAFLD development and progression, and severe injury in FPR2-depleted female mice supports the protective effect of FPR2 in the livers of female mice. Estrogen directly regulates FPR2 expression, which is related to estrogen-mediated protection against NAFLD. Therefore, our results suggest that FPR2 is a key target for understanding sex differences in NAFLD/NASH and has therapeutic potential for the prevention and treatment of NAFLD.

## Methods

**Animal experiments.** Male and female C57BL/6 mice (wild-type; WT) were purchased from Hyochang (Dae-gu, Korea). Fpr2-knockout (KO) mice derived from a mixed 129 S/SvEv x C57BL/6 strain[50] were generously donated from Dr. Kim (Pusan National University School of Medicine, Yangsan, Korea). All mice were housed with 12 h light/dark cycle and allowed free access to normal food and water at an average temperature of 22 °C ± 1 °C and humidity of 50 ± 10%. To mimic human-like NAFLD, 7-week-old male and female WT and KO mice fed normal chow-diet (Chow; M-diet; Optipharm.CO.,LTD, Cheongju, Korea) or choline-deficient, L-amino acid-defined high-fat diet (CDAHFD; A06071302; Research diet, New Brunswick, NJ, USA) ($n = 5$/group) consisting 60 kcal% and 0.1% methionine for 6, 12, and 36 weeks, respectively ($n = 4$/each chow group; $n = 5$/each CDAHFD group). At the end of each time point, mice were sacrificed to collect blood and liver samples. To assess the protective effects of estrogen-mediated Fpr2 in vivo, 6-week-old WT male mice received either placebo (P) ($n = 10$) or E2 pellets (E2; 0.36 mg; Innovative Research of America, Sarasota, FL, USA) ($n = 10$) in the mid-ventral subcutaneous region. One week after implementation, WT male mice were divided into randomly four experimental groups and fed Chow or CDAHFD for 12 weeks: Chow-P ($n = 4$), Chow-E2 ($n = 4$), CDAHFD-P ($n = 6$), and CDAHFD-E2 ($n = 6$). At the end of each time point, mice were sacrificed to collect blood and liver samples. To confirm whether Fpr2 is involved in protective function of estrogen in female mice, 5-week-old WT female mice underwent sham-surgery (Sham; $n = 9$) or ovariectomy (OVX; $n = 9$) as a surgical menopause model. Briefly, under isoflurane anesthesia, bilateral skin and muscle small incisions were made parallel to the dorsolateral of the midline. Bilateral ovaries were removed without disturbing the uterus, oviduct, and gonadal parametrial white adipose tissue. Then, the incision site was closed with a suture. In the Sham group, the abdominal wall was opened, and the ovaries were exteriorized to create similar stress, but not removed. After the 2 weeks recovery period, these female mice were divided into randomly four experimental groups and fed Chow or CDAHFD for 12 weeks: Chow-Sham ($n = 4$), Chow-OVX ($n = 4$), CDAHFD-Sham ($n = 5$), and CDAHFD-OVX ($n = 5$). At the end of each time point, mice were sacrificed to collect blood and liver samples. In addition, E2 pellets are given to female mice receiving OVX to double check the action of estradiol-mediated Fpr2 in the liver. Briefly, female mice were treated with estradiol (OVX-E2) ($n = 10$) or placebo pellet (OVX-P) ($n = 10$) at 1 week after they (5-week-old) underwent ovariectomy. Post 1 week after supplementation, these female mice were divided into randomly four experimental groups and fed Chow or CDAHFD for 12 weeks: Chow-OVX-P ($n = 4$), CDAHFD-OVX-P ($n = 6$), Chow-OVX-E2 ($n = 4$), and CDAHFD-OVX-E2 ($n = 6$). At the end of each time point, mice were sacrificed to collect blood and liver samples. Animal care and surgical procedures were approved by the Pusan National University Institutional Animal Care and Use Committee and carried out in accordance with the provisions of the National Institutes of Health Guide for the Care and Use of Laboratory Animals (Approval Number PNU-2020-2574 and PNU-2020-2641).

**Isolation of pHEPs and cell experiments.** To isolate primary hepatocytes (pHEPs), we performed a two-step collagenase perfusion according to the protocol of Selgen et al.[51]. Briefly, mice were anesthetized with isoflurane to immobilize them in the recumbent position on a treatment table, and the inferior vena cava was cannulated under aseptic conditions. Livers were perfused in situ with EGTA and collagenase (Sigma-Aldrich, St. Louis, MO, USA) to disperse the cells. Primary hepatocytes were separated from nonparenchymal cells using Percoll density gradient centrifugation. As determined by Trypan Blue exclusion, cell viability was >92% in all experiments. Primary hepatocytes were cultured on collagen-coated 6-well plates at a density of $1 \times 10^5$ cells / well or 60 mm plates at a density of $4 \times 10^5$ cells/plate in Williams' Medium E without phenol red (Sigma-Aldrich), supplemented with 5% fetal bovine serum (FBS; Gibco, Thermo Fisher Scientific, Waltham, MA, USA), 1 μM dexamethasone, and a cocktail solution of penicillin/streptomycin (P/S), ITS+ (insulin, transferrin, selenium complex, BSA, and linoleic acid), GlutaMAX™, and HEPES (Gibco). After cell attachment (~4 h after plating), the culture medium was replaced with serum-free Williams' Medium E containing 0.1 μM dexamethasone, and a cocktail solution of P/S, ITS+, GlutaMAX™, and HEPES for hepatocyte maintenance.

To assess the effect of Fpr2 in hepatocytes, pHEPs isolated from WT and KO female mice were serum-starved for overnight, and then were exposed to 250 μM of palmitic acid (PA; P0500; Sigma-Aldrich), a lipotoxicity-inducible agent, for 24 h. To assess whether Fpr2 induction by estradiol prevents hepatocyte damage from lipotoxicity, pHEPs from WT and KO male mice were cultured in a medium containing either vehicle (0.1% ethanol) or 100 nM E2 (E8875; Sigma-Aldrich) for 24 h to induce Fpr2 expression, and the cell medium was then changed after washing with PBS. These cells were treated with PA (250 μM) or 5% bovine serum albumin (BSA) for 24 h. To examine whether ligand-induced FPR2 protected hepatocytes against lipotoxic stress, pHEPs isolated from WT male mice were cultured with either vehicle (0.1% ethanol) or 20 nM Lipoxin A4 (LXA4; 90410; Cayman Chemical, Ann Arbor, MI, USA) for 24 h. After the cells were washed with PBS, these cells were treated with 250 μM PA or 5% BSA for 24 h to induce lipotoxicity.

These experiments were repeated at least three times.

**Analysis of human microarray data.** Human microarray data set were obtained from publicly available data sets from the Gene Expression Omnibus (GEO). The employed samples were 13 from healthy control (male; $n = 5$, female; $n = 8$) and 22 from patients with NAFLD (male; $n = 6$, female; $n = 16$) in adolescence (GEO access number: GSE66676)[20]. For the comparison between NAFLD patients and healthy control groups, the expression of *FPR2* with a fold change of 2.0 and adjusted *P*-value of <0.05 was considered differentially expressed, and the expression values were graphed as fold increase for healthy male.

**Cloning of vector constructs.** Putative binding sites for ERα, called estrogen response elements (EREs), within the promoter region of mouse or human FPR2 were analyzed by TRNASFAC database (version 8.0, geneXplain, Wolfenbüttel, Germany). To amplify the promoter region containing EREs, genomic DNA (gDNA) was isolated from mouse liver tissues and its concentration and purity were examined using Nanodrop (Thermo Scientific). The promoter region of Fpr2 containing EREs were amplified by PCR using the following primers: Forward, 5'-TTT GCT AGC TCT AAC CTG CAG GAT GGT CTG -3' and reverse, 5'-TTT TCT CGA GGC TTT GGC AGC AAA ATT CA-3'. The PCR product was purified using QIAquick PCR Purification Kit (Qiagen, Hilden, Germany), cut by the restriction enzymes, Nhe1 (R1031S; NEB, Ipswich, MA, USA) and Xho1 (R1046S; NEB), and then cloned into the pGL3 vector (Promega). The vector constructs with promote region of Fpr2 were transformed into competent E. coli (DH5α; HIT Competent Cells; RBC Bioscience; New Taipei City, Taiwan) and then plasmid DNA was extracted from well-transformed, ampicillin-resistant E. coli, using an AccuPrep Plasmid Mini Extraction Kit (Bioneer, Daejeon, Korea). The sequences of inserted regions were confirmed by sequencing analysis (Macrogen). Mutant vectors lacking the EREs were manufactured by Enzynomics (Daejeon, Korea) using the following primers: Forward, 5'- ATA GGG AGC TGA GTC GTG TTT GAT GTA GGC-3' and reverse, 5'- AAC TCC AGA TCC AAG GAC ACT AGA ATT CTA TTC C-3'. To identify whether the human FPR2 gene was also regulated by estrogen, vectors with wild-type human ERE or mutated human ERE was constructed by Enzynomics (Daejeon, Korea); following primers: wild-type-ERE vector forward, 5'- CTT ACG CGT GCT AGC GGC ATT ACC ATT CAG GAC ATA GGC-3' and wild-type-ERE vector reverse, 5'- ATC GCA GAT CTC GAG CCC CCA CCC CAC AAC AGT CCC CAG AGT-3'; mutant-type-ERE vector forward, 5'-AGT CGC AGC CAT CCC ATT ACT GGG TAT ATA CCC AAA G-3' and mutant-type-ERE vector reverse, 5'- AAT GGT ATT TCT AGT TCT AGA TCC CTG AGG AAT CAC-3'.

**Luciferase reporter assay.** AML12 cells (CRL-2254; ATCC, Manassas, VA, USA) or HepG2 cells (provided by Dr. Kim, Pusan National University, Pusan, Korea) cultured on 24-well plates at a density of $5 \times 10^4$ cells / well in Dulbecco's modified Eagle's medium (DMEM)/F-12 (Gibco) or DMEM without P/S. Using lipofectamine 2000 (Invitrogen) were transfected with pGL3 vector constructs for 24 h, and

then incubated with 100 nM estradiol (Sigma-Aldrich) or vehicle (0.1% ethanol) for 24 h. Next, these cells were harvested and tested with the Dual-Luciferase Reporter Assay System (Promega) following Manufacturers' instructions. All firefly luciferase activity data are normalized to Renilla luciferase activity.

**Cell proliferation assay.** Cell proliferation was measured with a Cell Titer Proliferation Assay (MTS; Promega) according to the manufacturer's instructions. In brief, pHEPs at a density of $4 \times 10^3$ cells/well were plated in 96-well plates and treated with either vehicle or 100 nM E2 for 24 h and the cell medium was then changed after washing with PBS. These cells were exposed to 250 μM of PA for an additional 24 h. After treatment, 10 μl of MTS reagent was added to each well, and the plates were incubated in 37 °C in a $CO_2$ incubator until the color developed. Absorbance was measured at the wavelength of 490 nm using a Glomax multi-detection system (Promega).

**Measurement of reactive oxygen species.** Reactive oxygen species (ROS) levels were determined using 2′,7′-Dichlorofluorescin diacetate (DCFH-DA, D6883, Sigma-Aldrich), following the manufacturers' instructions. Briefly, pHEPs were seeded into 96-well plates at a density of $4 \times 10^3$ cells/well. Then, pHEPs were treated with vehicle or E2 for 24 h. Following treatment with PA for an additional 24 h, cells were washed with ice-cold PBS and incubated with 10 μM of DCFH-DA for 30 min at 37 °C, protected from the light. Cells were washed three times with ice-cold PBS. Fluorescence intensity was detected with 488 nm excitation and 526 nm emission wavelengths by using a Glomax multi-detection system (Promega).

**Liver histology and immunohistochemistry.** To examine hepatic morphology and assess liver fibrosis, liver specimens were fixed 10% neutral buffered formalin, embedded in paraffin, and cut into 4 μm sections. Specimens were deparaffinized, hydrated, and stained as usual method with standard hematoxylin and eosin staining (H&E) and Sirius red staining as previously described[52]. For immunohistochemistry (IHC), sections were incubated for 10 min in 3% hydrogen peroxide to block endogenous peroxidase. Antigen retrieval was performed by heating in 10 mM sodium citrate buffer (pH 6.0) for 10 min or incubating with 0.2% pepsin for 10 min. Sections were blocked in protein blocking solution (X9090; Dako, Carpinteria, CA, USA) for 30 min and incubated with primary antibodies, active Caspase-3 (AF835; R&D Systems, Minneapolis, MN, USA), F4/80 (ab6640; Abcam, Cambridge, MA, USA), Fpr2 (ab203129; Abcam), CD68 (ab125212;Abcam) or non-immune sera to demonstrate staining specificity at 4 °C overnight. Polymer horseradish peroxidase (HRP) anti-rabbit (K4003; Dako) or HRP anti-rat IgG (A110-105P; BETHYL, Montgomery, Texas, USA) was used as the secondary antibody. 3,3'-Diaminobenzidine (DAB) (K3466; Dako) was employed for the detection procedure.

**Cell quantification.** For quantification analysis of Caspase-3, Fpr2, F4/80, or CD68,10 randomly chosen 40X fields/section were evaluated for each mouse. The positive cells for Caspase-3, Fpr2, F4/80, or CD68 were quantified by counting the total number of positive cells/field and dividing by the total number of hepatocytes/field for each mouse.

**Immunofluorescent staining.** For immunofluorescent staining, pHEPs and Kupffer cells isolated from WT and KO male/female mice were harvested and deposited in a monolayer onto a defined area of a slide by cytospin centrifugation (Cell spin; Hanil Scientific, Gimpo, Korea). Isolated cells were fixed in and permeabilized with cold acetone and methanol, respectively. These cells washed with TBS were incubated with blocking solution (X9090; Dako) for 30 min, and were treated with primary antibody, anti-Fpr2 (ab203129; Abcam) for at 4 °C overnight. After being washed with TBS, they were incubated with fluorescein-labeled secondary antibody, Alexa Fluor 568 goat anti-rabbit IgG (diluted 1:100; Invitrogen) for 30 min at room temperature. For double immunofluorescent staining, pHEPs stained for Fpr2 were further incubated with blocking solution for 10 min and then treated with secondary primary antibody, anti-albumin (sc-69873; Santacruz) for 2 h at room temperature. After being washed, their pHEPs were incubated with fluorescein-labeled secondary antibody, Alexa Fluor 488 chicken anti-mouse IgG (diluted 1:100; Invitrogen) for 30 min at room temperature. Slides were mounted on slides antifade mounting medium with 4′,6-diamidino-2-phenylindole (DAPI, VectaShield, Burlingame, CA, USA). Slides were viewed with a Zeiss LSM 800 confocal microscope (Carl Zeiss Inc., Thornwood, NY, USA).

**Oil Red O staining.** Staining of pHEPs was performed as previously described. Cells were fixed with 4% paraformaldehyde in PBS for 10 min. After fixation, the cells were washed with glass-distilled $H_2O$ three times for 30 sec per rinse. Next, cells were incubated with 100% propylene glycol (PEG, Sigma-Aldrich) for 2 min then stained with Oil Red O (0.5% in PEG, Sigma-Aldrich) revealing lipid droplets. Following incubation of staining solution for overnight, the wells were washed in 60% PEG for 1 min. Nuclei was counterstained with hematoxylin for light microscopic examination.

**Hepatic triglyceride assay.** Hepatic triglyceride (TG) levels were measured using Triglyceride Colorimetric assay kit (Cayman Chemical), following the manufacturers' instructions. Briefly, liver tissue was homogenized using NP40 and tissue homogenates were centrifuged at 10,000 r.c.f. for 10 min at 4 °C. The supernatants containing TG were used subsequent biochemical analysis. 10 μl of supernatants and 150 μl of enzyme solution were added to each well and then incubated for 15 min. Absorbance was measured at the wavelength of 540 nm using a Glomax multi-detection system (Promega).

**Serum and hepatic biochemical analysis and enzyme-linked immunosorbent assay.** Serum aspartate aminotransferase (AST/GOT, glutamate-oxaloacetate transaminase) and alanine aminotransferase (ALT/GPT, glutamate pyruvate transaminase) were measured using GOT reagents (AM103-K; Asan Pharmaceutical, Seoul, Korea) and GPT reagents (AM102-K; Asan Pharmaceutical) according to the manufacturer's instructions.

Serum levels of 17β-estradiol, TNF-α, and IL-6 were measured using a mouse ELISA kit of 17β-estradiol (ES180S-100; Calbiotech, El Cajon, CA, USA), IL-6 (M6000B; R&D systems), TNF-α (MTA00B; R&D system) according to the manufacturer's instructions.

The levels of hepatic LXA$_4$ were measured using mouse LXA$_4$ ELISA kit (CSB-E13268m, Cusabio, Wuhan, China) following to manufacturer's protocols.

**Western blot assay.** Total protein was extracted from primary cells or freeze-clamped liver tissue samples that had been stored at −80 °C. Samples were homogenized in Triton lysis buffer (TLB) supplemented with protease inhibitor (Complete Mini; Roche, Indianapolis, IN, USA) and centrifuged at 13,000 x g. for 15 min at 4 °C. The supernatants containing protein extracts were used in subsequent biochemical analysis. Protein concentration was measured with a Pierce BCA Protein Assay kit (Thermo Scientific). Before the pooling of protein lysates, individual protein expression was confirmed (data not shown). Then, equal amounts of protein lysates from representative mice ($n = 3$) per treatment group were combined and protein concentration was measured. To denature and reduce protein samples, proteins were boiled in 5X sample buffer containing β-mercaptoethanol and sodium dodecyl sulfate (SDS) at 100 °C for 10 min. Total 50 μg of protein lysates was separated by SDS-polyacrylamide gel electrophoresis (PAGE) on 10 or 12% tris-glycine gel and transferred onto a 0.45 μm pore size polyvinylidene difluoride (PVDF) membrane (Millipore, Darmstadt, Germany). Primary antibodies against rabbit polyclonal anti-Fpr2 (diluted 1:1000; NLS1878; Novus Biologicals), mouse monoclonal anti-α-Sma (diluted 1:1000; A5228; Sigma-Aldrich), rabbit polyclonal anti-Col1α1 (diluted 1:1000; NBP1-30054; Novus Biologicals, LLC, USA), rabbit polyclonal anti-Cleaved caspase-3 (diluted 1:1000; 9661; Cell signaling technology, Danvers, MA, USA), polyclonal anti-Caspase-3 (diluted 1:1000; 9662; Cell Signaling), mouse monoclonal anti-Glyceraldehyde 3-phosphate dehydrogenase antibody (Gapdh; diluted 1:1,000; AbD Serotec, Oxford, UK) were used in this experiment. HRP-conjugated anti-rabbit or anti-mouse IgG (Enzo Life Sciences, Inc., Farmingdale, NY, USA) was used as secondary antibody. Protein bands were detected using an EzWestLumi ECL solution (ATTO Corporation, Tokyo, Japan) as per the manufacturer's specifications (ATTO Corporation, Ez-Capture II). Band intensities were calculated using the CS analyzer 4.0 program (Version1.0.3, ATTO).

**RNA analysis.** Total RNA was extracted from liver tissue or pHEPs by using Trizol reagent (Invitrogen). The concentration and purity of RNA were determined using nanodrop. Template complementary DNA was synthesized from total RNA using the SuperScript First-strand Synthesis System (Invitrogen) according to manufacturers' instructions. We performed the real-time qRT-PCR analysis by using Power SYBR Green Master Mix (Applied Biosystem) on the manufacturers' specifications (QuantStudio 1; Thermo Scientific). All reactions were duplicated, and data were analyzed according to the ΔΔCt method. The expression values were normalized to the levels of mouse 40 S ribosomal protein S9 mRNA. The sequences of all primers used in this study are summarized in Supplementary Table 1. All PCR products were directly sequenced for genetic confirmation (Macrogen).

**Hydroxyproline assay.** Hydroxyproline content of the livers was calculated by the method previously described[53]. Briefly, 50 mg of freeze-dried liver tissue was hydrolyzed in 6 N HCl at 110 °C for 16 h. The hydrolysate was evaporated under vacuum and the sediment was re-dissolved in 1 ml distilled water. Sample were filtered using 0.22 μm filter centrifuge tube (Corning Incorporated, Corning, NY, USA) at $16,873 \times g$. for 5 min. 0.5 ml of chloramine-T solution, containing 1.41 g of chloramine-T dissolved in 80 ml of acetate–citrate buffer and 20 ml of 50% iso-propanol, were added and incubated at room temperature for 20 min. Then, 0.5 ml of Ehrlich's solution containing 7.5 g of dimethylaminobenzaldehyde dissolved in 13 ml of 60% perchloric acid and 30 ml of isopropanol, was added to the mixture and incubated at 65 °C for 15 min. After cooling at room temperature, the standard and samples were measured by a spectrophotometer at 561 nm. Amount of hydroxyproline in each sample was determined using a regression curve from high purity Trans-4-hydroxy-L-proline (Sigma-Aldrich) as a standard. Total hydroxyproline was calculated based on individual liver weights (mg hydroxyproline/mg liver).

**Statistics and reproducibility**. Results are expressed as mean ± S.E.M. Statistically significant differences between the control and treatment groups or subgroups were analyzed with two-tailed unpaired Student's $t$-test, two-way analysis of variance (ANOVA) followed by post hoc Tukey's test. Differences were considered as significant when $P$-values were <0.05. The degree of correlation between the expression level of hepatic Fpr2 and concentration of serum estradiol was analyzed by Pearson's correlation coefficient. Statistical analyses were performed using IBM SPSS Statistics 21 software (Release version 21.0.0.0, IBM Corp., Armonk, NY, USA) and GraphPad Prism 8 (GraphPad Software Inc.). All experiments were repeated independently at least three times and no data were excluded from the analyses. For all animal experiments, mice were randomly assigned to different experimental groups, and investigators were blinded to the groups during the data collection and analysis. Exact information on the sample numbers being analyzed can be found in Supplementary Data.

**Reporting summary**. Further information on research design is available in the Nature Research Reporting Summary linked to this article.

## Data availability

Source data are provided with this paper. The datasets used in this study are available in the GEO database under accession code GSE66676[20]. The remaining data are available within the article, Supplementary Information, Supplementary Data, or Source Data file. Source data are provided with this paper.

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

## Acknowledgements

This work was supported by Youngmi Jung and the National Research Foundation (NRF) of Korea funded by the Korean Government (MSIT) (2018R1A2A3075038) to Y.J.. We greatly appreciate Anna Mae Diehl's critical comments on the research.

## Author contributions

Y.J. conceived and designed the study; C.L., J.H., D.O., M.K., and H.J. carried out experiments; C.L., J.K., T.-J.K, S.-W.K., J.N.K., Y.-S.S., J.H.K., A.S., and Y.J. analyzed data; C.L., J.K., and Y.J. wrote the manuscript Y.J. obtained funding. All authors read and approved the manuscript.

## Competing interests

The authors declare no competing interests.
