## [Peer Review File · Nature Communications]

Reviewers' Comments:

Reviewer #1:

Remarks to the Author:

This manuscript by Lee and colleagues evaluates the role of Formyl peptide receptor 2 on sex-specific progression of NAFLD/NASH. The study of NAFLD and NASH are particularly relevant given the increased prevalence around the world. The paper is difficult to follow at times and has instances of grammatical errors throughout. The authors utilized a single model of liver disease in this report (choline-deficient/amino-acid specific diet) and omitted several quantitative methods and appropriate control and/or confirmatory groups. The number of animals evaluated for each experiment is also very low. Work from our laboratory has shown that liver disease phenotypes vary greatly from mouse to mouse, as does their endogenous hormone levels. This is why higher N's and transparent quantitative methods are requested from this specific reviewer. My concerns are as follows:

1. Mice do not identify with a specific gender, I suggest changing all instances throughout the manuscript to say sex-specific as opposed to gender-specific.
2. Figure 1D: Its unclear if the active Caspase-3 IHC was quantified.
3. Figure 2C-Sirius Red: Again, its unclear if the collagen deposition is quantified. Also, this needs to be done by using a slide-scanner to capture the entire cross-section and then quantified using an imaging algorithm.
4. Figure 2C-F4/80: Is this quantified? Also, F4/80 can detect a myriad of immune cells, not just Kupffer cells in the liver. In fact, NAFLD and NASH are closely associated with the recruitment of monocytes/macrophages into the liver. You are not only measuring Kupffer Cells here.
5. These 17beta-E2 levels seem low. The 36 week treatment group is still just under 10 months of age. It would have been nice to see if a 17beta-E2 recovery dose increased the expression of Fpr2.
6. Figure 3E: 9 samples gave a really strong correlation here. I would take multiple other sets of mice to confirm this.
7. The number of mice per experimental group throughout the manuscript is strikingly low.
8. Why was the FPR2 data from KO mice not shown in the supplement?
9. Supplemental Figure 7 & Figure 9: Why was this experiment not performed in WT & hepatocyte-specific ERaKO male mice? This could have directly supported your culture data? Also, the dose of 17Beta-E2 provided seems exceedingly high based on your data sown in Figure 3D.
10. The OVX female study should have also implemented a 17Beta-E2 recovery group. There is a fair amount of data now showing that ovarian factors other than 17Beta-E2 are likely altering systemic health parameters including metabolism. The inclusion of the replacement 17Beta-E2 group would have provided additional evidence that your effects (in very low N's) are indeed related to 17Beta-E2.
11. The paper could benefit dramatically from some human data showing how estrogen status correlates with hepatic FPR2 expression/activity.

Reviewer #2:

Remarks to the Author:

Lee et al present an interesting study where a non-redundant role for mouse Fpr2 is shown in the context of NAFLD, with an interesting sex regulation. In itself the study is well presented, though repetitive in part (same assays and data presentation for each therapeutic or KO or hormone replacement modality), but I think there are still major uncealr points which detract from the

overall impact of the data presented herein.

Major ones:

1. What are the ligands for Fpr2 which are necessary to afford hepato-protection?
2. Is the relevant Fpr2 agonist controlled in a sex hormone-related manner? or is the regulation of estrogens centred only on receptor expression.
3. Some the data presented are not congruous, with significant differences - in some case - be evident at 36 hour time-point and other at different time points. This could be related to the different readouts under study. However, the Fpr2 expression does not seem different between males and females (Fig 3a and 3b).
4. The statistical assay applied is wrong. Many of the experimental designs have two variables, e.g. males and females, which change over time, e.g. 6, 12 and 36 weeks. Such designs would require a two-way ANOVA analysis to identify potential statistical differences.

Other ones:

1. Introduction (Page 1, Lines 46-50). What is the quantitative incidence of NALFD in man, pre-menopausal women, and post-menopausal women. Good to give an idea of the numbers.
2. The word sex should be used instead of gender, which has more a societal connotation (e.g. Results, Page 1, Line 77). This is biology hence the focus is on sex differences.
3. If there is a sex control on Fpr2 (which I find quite unlikely from the data presented) I would have expected differences to become evident at a earlier time-point and not 36 weeks (e.g. Results, Page 1, Line 95). By then there could be several intermediate mechanisms between sex hormones, liver inflammation, and outcome of NALFD.
4. Which are the cell types expiring Fpr2 in the liver? How come the IHC expression seems to fade with ageing, e.g. female at 36 week (e.g. Results, Page 1, Line 143-144) when the only occasional statistical significance seem to be apparent at week 36? Unless the application of two-way ANOVA does not reveal different statistical differences. These two facts do not seem to be congruent.
5. (e.g. Results, Page 2, Line 151). Is Fpr2 truly highly expressed in female livers? Fig. 3b would need some semi-quantitative analyses from several mice in order to have an idea of protein expression. IHC would need to identify the cells that express Fpr2.
6. (e.g. Results, Page 4, Line 238). What is MT ERE?
7. Figures where time-courses and sexes are run over 36-week should highlight potential differences among the two sex groups, not compare the effect of time, as one presume that the disease develops over time. In some cases there are too many comparison, many of which not that relevant, e.g. Fig. 1b, Fig.5c and more.
8. Similarly, many of the Figures present IHC data on representative images. It could be important to provide some quantitative data on images from different mice to understand the spread, and values, of the data.
9. Figure 9a: it is strange that there is no difference in estradiol levels between the two diets? If so, how could one link pathological changes associated with the diets to Fpr2 control by sex hormones? Fig 9 seems to indicate that the changes in Fpr2 expression are not dependant from serum estradiol levels. Apology if I misunderstood this point.

Reviewer #3:

Remarks to the Author:

The MS entitled "Formyl peptide receptor 2 determines sex-specific differences in the progression of nonalcoholic fatty liver disease/steatohepatitis" and authored by Chanbin Lee et al described that the presence of Formyl peptide receptor 2 regulated by estradiol in female mice protect from the development and progression of NAFLD.

Although the authors showed appealing results, the MS in the present form is quite descriptive.

The authors need to identify the mechanism for which Formyl peptide receptor 2 expression in the liver boost the reduction of TG content.

Major points:

-First the authors should determine the levels of Formyl peptide receptor 2 in healthy human liver from male and female. RNA seq or arrays expression are already available in the literature. Also the expression of Formyl peptide receptor 2 should be analyzed in NAFLD patients with different sex.

- Also, it is necessary to determine if the promoter of the human gene Formyl peptide receptor 2 is also a target for estradiol or this is specific for the gene belong to the mice.

- In figure 2 a deeper analysis should be performed. A more detailed profile of inflammatory response is required, as well as the evaluation of different hormones. In the MS, the authors should evaluate the WAT and the BAT as well as the glucose metabolism. Additionally, food intake should be measured in these animals as well as the body weight during the timing 6, 12 and 36 weeks.

Moreover, DNL, beta oxidation and VLDL secretion should be evaluated in these mice.

- In fig 3 the authors show changes in estradiol at 6 weeks, but no changes were detected at 12 and 36 weeks. However, Formyl peptide receptor 2 is express at different levels at 3, 12 and 36 hours with a positive tendency in comparison to the male mice. How the authors explain this lack of correlation between estradiol and Formyl peptide receptor

-Same approaches should be taken in fig 4 to determine inflammatory response, hormones, WAT and the BAT characterization, glucose metabolism, food intake as well as the body weight during the timing 6, 12 and 36 weeks. Also the mechanism underlying the regulation of lipid content in the KO mice is required to be analyzed. It has been previously reported the effect of Formyl peptide receptor 2 in the inflammatory response, but the TG content in the liver at 6 weeks is already different between male and female. The mechanisms underlying this effect should be analyzed and include in the present MS.

-In fig 8 in primary hepatocytes lipid context, mitochondrial ROS, inflammatory response and the mechanism underlying the lack of Formyl peptide receptor 2 should be evaluated.

- Finally, in Fig 10, it will be relevant to identify if estradiol treatment will increases the levels of Formyl peptide receptor 2 in OVX mice.

Reviewer #1 (Remarks to the Author):

This manuscript by Lee and colleagues evaluates the role of Formyl peptide receptor 2 on sex-specific progression of NAFLD/NASH. The study of NAFLD and NASH are particularly relevant given the increased prevalence around the world. The paper is difficult to follow at times and has instances of grammatical errors throughout. The authors utilized a single model of liver disease in this report (choline-deficient/amino-acid specific diet) and omitted several quantitative methods and appropriate control and/or confirmatory groups. The number of animals evaluated for each experiment is also very low. Work from our laboratory has shown that liver disease phenotypes vary greatly from mouse to mouse, as does their endogenous hormone levels. This is why higher N's and transparent quantitative methods are requested from this specific reviewer. My concerns are as follows:

1. Mice do not identify with a specific gender, I suggest changing all instances throughout the manuscript to say sex-specific as opposed to gender-specific.

: We appreciate your helpful comment for improving the quality of our manuscripts. As your comment, we changed "gender" with "sex" in the revised manuscript.

2. Figure 1D: Its unclear if the active Caspase-3 IHC was quantified.

: In the revised manuscript, we counted the number of the active caspase-3 positive cells and graphed it (Fig. 1e), as you requested. We referred these articles in cell quantification [Gut. 2010 May;59(5):655-65./ Gastroenterology. 2008 May;134(5):1532-43./ Biomaterials. 2019 Oct;219:119375.]. The method was described in the revised manuscript.

3. Figure 2C-Sirius Red: Again, its unclear if the collagen deposition is quantified. Also, this needs to be done by using a slide-scanner to capture the entire cross-section and then quantified using an imaging algorithm.

: As you know, Sirius red staining shows the morphological collage deposition in the liver without quantification information. Hence, we measured hepatic hydroxyproline content in all mice to quantify liver fibrosis. Hepatic hydroxyproline assay is an established biochemical measurement of liver fibrosis [Nat Commun. 2016;7:13817./ Nat Commun. 2020;11(1):2362./ Hepatology. 2020;10.1002/hep.31604./ J Hepatol. 2021;74(3):638-648./ Gastroenterology. 2019;157(3):777-792.e14./ Gut. 2010;59(5):655-65./ Biomaterials. 2019 Oct; 219: 119375/ Nat Nanotechnol. 2021 Apr;16(4):466-477./ Nat Commun. 2016 Mar 22;7:10993.].

4. Figure 2C-F4/80: Is this quantified? Also, F4/80 can detect a myriad of immune cells, not just Kupffer cells in the liver. In fact, NAFLD and NASH are closely associated with the recruitment of monocytes/macrophages into the liver. You are not only measuring Kupffer Cells here.

: Even though F4/80 is a representative surface glycoprotein and expressed highly by various macrophages including Kupffer cells, F4/80 is used in assessing Kupffer cells in many articles [Nat Commun. 2021;12(1):213./ Nat Commun. 2014; 5:3862. Hepatology. 2010;51(2):511-22. / Nutrients. 2019;11(4):857.]. Hence, we used F4/80 to detect Kupffer cells in the research. However, we agreed with your point. Therefore, we additionally examined expression of CD68, a marker of Kupffer cells [Hepatology. 2012; 56(2):735-46./ J Cell Sci 1987; 87: pp. 113-119./ J Hepatol. 2010; 53(5):903-10.].

We also provided quantitative analysis of F4/80 and CD68-positive cells in the revised manuscript (Fig. 2d). Quantification analysis confirmed the significant increase of F4/80 or CD68-positive cells in male mice than female mice.

5. These 17beta-E2 levels seem low. The 36 week treatment group is still just under 10 months of age. It would have been nice to see if a 17beta-E2 recovery dose increased the expression of Fpr2.
: The 36 week groups were treated with a diet for 36 weeks, and these mice at the 36W groups are 43-week old (7w+36w), not under 10 months of age. 10~14-month-old mice correspond to the middle age of humans (38-47 years), which is the perimenopause period at which estradiol levels gradually decrease [*Okajimas Folia Anat. Jpn.* **65**, 35-42 (1988)]. We used mouse estradiol ELISA kit produced by Calbiotech. Other groups using the same kit showed the similar level of serum estradiol with ours [Nat Commun. 2019 Dec 17;10(1):5745./ Am J Physiol Lung Cell Mol Physiol. 2019; 317(5):L702-L716./ Am J Physiol Endocrinol Metab. 2015;308(12):E1066-75./ J Neurosci. 2021;41(4):648-662.].

We showed that the elevated estradiol level by estradiol supplement increased Fpr2 expression in male mice, and the alleviated estradiol amount by ovary removal decreased Fpr2 level in female mice in Fig. 9 and 10. Male mice-isolated hepatocytes treated with estradiol also upregulated Fpr2 expression (Fig. 8). These data present that estradiol impacts Fpr2 expression. In addition, we checked whether estradiol supplement in OVX-female mice could recover Fpr2 expression in the revised manuscript (Supplementary Fig. 15-16), as you requested; **“To double check the action of estrogen-regulated Fpr2 expression in NAFLD, ovariectomized WT female mice were supplemented estradiol (OVX-E2) or placebo pellets (OVX-P), then were fed chow or CDAHFD for 12 weeks (Supplementary Fig. 15a). Reduced levels of serum estradiol and hepatic Fpr2 expression significantly increased in the OVX-E2 compared with the OVX-P groups (Supplementary Fig. 15f and 16a-c). Compared with the CDAHFD-fed OVX-P group, the CDAHFD-treated OVX-E2 group had the reduced liver damage and accumulation of Caspase 3-positive cells (Supplementary Fig. 15b-d and 16d). Estradiol supplementation also mitigated the enhanced hepatic fibrosis and inflammation in CDAHFD-given female mice with OVX (Supplementary Fig. 15e-g and 16d-f). In the chow groups, hepatic injury by estradiol supplement were ascent (Supplementary Fig. 15-16).”**

6. Figure 3E: 9 samples gave a really strong correlation here. I would take multiple other sets of mice to confirm this.

: As you requested, we added more samples in analyzing correlation of *Fpr2* with estradiol level (4 mice/group) and got the more concredited data ($r=0.883$, $p<0.001$). However, male mice were not included in the analysis because *Fpr2* level in these mice is already too low.

7. The number of mice per experimental group throughout the manuscript is strikingly low.

: In these studies, we designed 36 groups and used almost 250 mice. In each group, we assigned 4-6 mice per group; 5-6 mice for the treatment group and 4 mice for the control group. Because the variation among animals in the control group is less than that in the treatment group, a small number (relatively) of animals is employed in the control group. The number of mice per group in our research is similar with that in other research groups [Nat Commun. 2021;12(1):66./ Nat Commun. 2020;11(1):5807./ Hepatology. 2019 Apr 25/ Hepatology. 2014;60(1):133-45.]. In addition, we chose

the representative 4 mice per group to assess gene expression by qRT-PCR because of the limited number of wells; 4 mice / chow group or 5-6 mice / CDAHFD group, 4 groups (two chow and two CDAHFD group), and duplicated reactions require 36-40 wells for an interested gene and 36-40 wells for the 9S (internal control). Hence, total 72-80 wells were occupied in examining the expression of one gene; one gene per one time running. Because of the limited number of wells in qRT-PCR, it is impossible to examine the gene expression of all mice. To compensate the limitation, all samples were analyzed in IHC.

In regard of comment on single animal model: Although we employed single diet model, we used KO mice and additional surgical models, such as estrogen supplements and OVX. Also, new additional animal experiment model which reviewer requested was included in the revised manuscript. Furthermore, female mice were employed in the research. In most of research papers using in vivo, male mice, not female mice, are employed. Even in vitro assays, primary cells were used. Compared with other researches, 36 groups and 235 mice are not small number. Considering the number of mice used in animal experiments throughout the manuscript, the number of mice per experimental group is not insufficient.

		NAFLD animal model						Cell Isolation
		6W		12W		36W		
		Chow	CDAHFD	Chow	CDAHFD	Chow	CDAHFD	
WT	male	4	6	4	6	4	5	20
	female	4	5	4	6	4	5	10
KO	male	4	6	4	6	4	5	20
	female	4	5	4	6	4	6	10

		Estradiol supplementation				Ovariectomy				E2 recovery in OVX			
		Placebo		Estradiol		Sham		OVX		Sham		OVX	
		Chow	CDAHFD	Chow	CDAHFD	Chow	CDAHFD	Chow	CDAHFD	Chow	CDAHFD	Chow	CDAHFD
WT	male	4	6	4	6	4	6	4	6	4	6	4	6
	female					4	6	4	6	4	6	4	6

Total	235
--------------	------------

8. Why was the FPR2 data from KO mice not shown in the supplement?

: We used systemic Fpr2-lacking mice and Fpr2 expression was not detected in whole liver and primary hepatocytes isolated from Fpr2 KO mice (Supplementary Fig. 6). Because too many data were already presented, we thought that we did not need to show images with nothing. For your convenience, we provide Fpr2 staining image in liver cells isolated from KO mice.

9. Supplemental Figure 7 & Figure 9: Why was this experiment not performed in WT & hepatocyte-specific ERaKO male mice? This could have directly supported your culture data? Also, the dose of 17Beta-E2 provided seems exceedingly high based on your data shown in Figure 3D.

: WT and hepatocyte-specific ERaKO mice is suitable to female mice, but not male mice. In female mice, estrogen should be suppressed, and hepatocyte-specific ERaKO is a proper model, as you point out. However, estradiol should be implanted into male mice to examine whether exogenous estradiol stimulates Fpr2 in male mice, because estrogen level is lower in male mice. To match the model, we used estradiol implantation for the estradiol supplementation and ovary removal for the estradiol suppression. Estrogen treatment increased Fpr2 expression in the liver of WT male mice having low estrogen concentration, thereby reducing liver damage caused by CDAHFD. Hepatic Fpr2 level was downregulated in females with OVX, indicating that OVX-female mice were more susceptible to liver damage caused by CDAHFD. These results sufficiently support that the estradiol-regulated Fpr2 has a hepatoprotective effect against the CDAHFD-induced liver damages. In addition, recovery experiments in OVX-female mice support these findings (please see the answer for comment #5).

In addition, to employ hepatocyte-specific ERaKO mice, we need to use double-KO mice (albumin/cre X ERaF/F) and choose female mice only from these mice; the probability of obtaining the double KO female mice needed in the experiments is 6.25% ($1/8 \times 1/2 = 1/16$). To set up the NASH model, at least 10 double KO female mice and 10 F/F female mice are needed. To get these mice, we have to breed at least 160 mice. In the present study, 36 animal groups and almost 250 mice were already used. The number of animals used is not a small number rather than a huge number. Considering overall findings in the research, the experiment of estrogen suppression is relatively minor. For this experiment, using many numbers of mice who are needed to get double KO female mice is a little immoderate. We hope you would generously accept our explanation.

In regard of the comment for E2 level in Fig.3, Fig. 9, and Fig. 10:

Fig. 3d presents serum estradiol levels in mice without exogenous estradiol treatment, and Fig. 9a shows them in mice with exogenous estradiol supplement. Hence, the P (placebo) groups in Fig. 9a are similar to the WT male groups in Fig. 3d. Considering the treatment period of a diet, the P groups shown in Fig. 9a can be compared with the 12W male groups in Fig. 3d, and estradiol levels among these mice are similar, not different. Likewise, the sham group in Fig. 10a can be compared with the 12W female groups in Fig. 3d. Based on the previous studies [Cancer Res. 2016 Oct 1;76(19):5657-5670./ Biochim Biophys Acta. 2015 Oct;1852(10 Pt A):2161-9/ J Mol Cell Cardiol. 2016 May;94:180-188./ J Endocrinol. 2006 Jun;189(3):519-28./ Lipids. 2014 Aug;49(8):745-56./ J Immunol. 2015 Mar 15;194(6):2522-30.], we implanted 0.36mg of E2 into mice.

10. The OVX female study should have also implemented a 17Beta-E2 recovery group. There is a fair amount of data now showing that ovarian factors other than 17Beta-E2 are likely altering systemic health parameters including metabolism. The inclusion of the replacement 17Beta-E2 group would have provided additional evidence that your effects (in very low N's) are indeed related to 17Beta-E2.

: As you requested, we conducted the additional experiments of E2 recovery in female mice with OVX. Before feeding diet, WT female underwent ovariectomy and took a recovery period of one

week (Supplementary Fig. 15). And then these mice received E2 or placebo pellet in the mid-ventral subcutaneous region. One week after E2 supplementation, these female mice were fed either chow or CDAHFD diet for 12 weeks. The explanation for the experimental method was added in the revised manuscript; **“In addition, E2 pellets were given to female mice receiving OVX to double check the action of estradiol-mediated Fpr2 in the liver. Briefly, female mice were treated with estradiol (OVX-E2) (n=10) or placebo pellet (OVX-P) (n=10) at one week after they (5-week-old) underwent ovariectomy. Post one week after supplementation, these female mice were divided into randomly four experimental groups and fed Chow or CDAHFD for 12 weeks: Chow-OVX-P (n=4), CDAHFD-OVX-P (n=6), Chow-OVX-E2 (n=4), and CDAHFD-OVX-E2 (n=6). At the end of each time point, mice were sacrificed to collect blood and liver samples.”**.

The description for the results was added in the revised manuscript (Supplementary. Fig. 15 and 16). Please see the answer for comment # 5.

11. The paper could benefit dramatically from some human data showing how estrogen status correlates with hepatic FPR2 expression/activity.

: As you commented, we analyzed hepatic FPR2 expression in human using published microarray data to investigate the potential role of FPR2 in its clinical implications. We wanted to analyze the relationship of hepatic FPR2 with serum estrogen concentration, but the data which we want to have were not available. There are two separated data sets, adolescence (GEO access number: GSE66676) and adult group (GEO access number: GSE86932). However, we could not use the adult group, because age range of healthy women is too broad and 3 out of 5 healthy females were in their early 40's, and it is ambiguous age to determine reproductive status and estrogen reduction. On the other hand, adolescence group have an increasing level of estrogen in girls compared with boys. Hence, we used the data of adolescence group in the analysis. The explanation for the results were added in the revised manuscript; **“To determine if a similar correlation might be detected in human, we analyzed hepatic *FPR2* expression in human using published microarray data (GEO access number: GSE66676). The age of human included in the cohort ranges from 13 to 19 and represents the group having an increasing level of estrogen in girls compared with boys. Hepatic *FPR2* expression was significantly higher in healthy females than healthy males (Supplementary Fig. 5), while its level was dramatically reduced in female patients with NAFLD and similar with it in male patients. There was no difference of *FPR2* expression between healthy men and male patients with NAFLD.”**.

Reviewer #2 (Remarks to the Author):

Lee et al present an interesting study where a non-redundant role for mouse Fpr2 is shown in the context of NAFLD, with an interesting sex regulation. In itself the study is well presented, though repetitive in part (same assays and data presentation for each therapeutic or KO or hormone replacement modality), but I think there are still major unclear points which detract from the overall impact of the data presented herein.

Major ones:

1. What are the ligands for Fpr2 which are necessary to afford hepato-protection?

: Since functions of FPR2 are different depending on types of agonists, experimental conditions, and so on, we choose the Lipoxin A4 (LXA₄), a well-known FPR2 ligand, based on previous researches in which LXA₄ showed the protective effects in various disease models such as acute liver failure, liver fibrosis and obesity [Int J Mol Med. 2016;37(3):773-80/ Turk J Gastroenterol. 2019; 30(8): 745–757/ Cell Metab. 2015; 22(1): 125–137]. We examined whether LXA₄ upregulated Fpr2 expression, and LXA₄-induced Fpr2 protected hepatocytes from lipotoxicity caused by PA. To determine the optimal concentration of LXA₄, 10, 20, or 100nM of LXA₄ was given to primary hepatocytes isolated from male mice, and 20nM was chosen because FPR2 was significantly upregulated in 20nM of LXA₄-treated cells compared with other treatment groups. And then, LXA₄ (20nM)-treated hepatocytes were exposed to PA (Please see below figure). Elevated *Fpr2* expression by LXA₄ was reduced by PA treatment, but its level was significantly higher in the LXA₄+PA group than the vehicle+PA group. In addition, the LXA₄+PA group had the increased *G6pc* expression and cell viability and the decreased apoptosis level compared with the vehicle+PA group. Without cell injury, LXA₄ hardly impacted hepatocyte responses, such as *G6pc* expression, cell viability, and apoptosis. As your comment, the data obtained from the ligand experiments support hepato-protected action of FPR2. However, the present study reveals the association of FPR2 with sex-specific NAFLD development and/or progression. If the ligand data would be included in the manuscript, it is reasonable to investigate the effect of LXA₄ in all experiments (all *in vivo* models) which we have conducted. It will be another huge experiments for LXA₄ functions, and generate the massive data. Also, we could not find the appropriate part to add the ligand data in the manuscript because it seems to digress from the main subject. Hence, we briefly described the effect of LXA₄ in hepatocyte protection from PA injury in the discussion section to support the FPR2 action; **“In addition, we employed FPR2 agonist, lipoxin A₄ (LXA₄), to examine whether LXA₄ upregulated FPR2 expression, and LXA₄-induced FPR2 protected hepatocytes from lipotoxicity caused by PA, and found that upregulated FPR2 by LXA₄ also protected hepatocytes from PA injury (data not shown)”**.

We present the data in the answer, for your convenience. However, if you request us to add the data in the manuscript, we will do it. If you advise us, it will be very helpful. Thanks.

LXA₄-induced Fpr2 protects hepatocytes against lipotoxicity

(a) A scheme for cell experiment in which pHEPs treated with either vehicle or 20 nM of LXA₄ were exposed to 250 μM of PA for 24 hours. (b) qRT-PCR for *Fpr2* and *G6pc*, (c) cell viability, (d) western blot analysis for Fpr2, cleaved Caspase-3, and pro Caspase-3 in these cells. Gapdh was used as internal control. Data shown represent one of three experiments with similar results.

2. Is the relevant Fpr2 agonist controlled in a sex hormone-related manner? or is the regulation of estrogens centred only on receptor expression.

: We have showed that Fpr2 upregulation triggered by estradiol protects hepatocytes from PA, suggesting that estradiol seems to center on Fpr2 expression. To find out whether estradiol influences expression of FPR2 agonist in hepatocytes, as you questioned, it should be assumed that hepatocytes express FPR2 agonist. As we answered above to the question about the FPR2 agonist, LXA₄ increased FPR2 expression, which protected hepatocytes from lipotoxicity. LXA₄ is known to mediate interaction of immune cells with various types of cells, such as osteoblast, platelet, and hepatocytes [BMC Musculoskelet Disord. 2009; 10: 57/FASEB J. 2002 Dec; 16(14):1937-9]. However, LXA₄ is not produced by hepatocytes, except one condition; aspirin-treated hepatocytes make aspirin-triggered 15-epi-LXA₄, which is involved in pharmacological action of aspirin [J Exp Med. 1997; 185 (9): 1693-1704 / Proc Natl Acad Sci U S A. 1995; 92 (21): 9475–9479.]. Kupffer cells, liver-resistant macrophage, are known to make both native LXA₄ and aspirin-triggered 15-epi-LXA₄ from endogenous arachidonic acid [Prostaglandins Leukot Essent Fatty Acids. 2005;73(3-4):277-82/FASEB J. 2002 Dec; 16(14):1937-9.]. Kupffer cells-derived LXA₄ reduced cytokine-induced chemostasis in adjacent hepatocytes [Front Immunol. 2012 Aug 20;3:257. /FASEB J. 2002;16(14):1937-9.]. Based on these findings, estrogen-mediated LXA₄ expression should be examined in Kupffer cells, not hepatocytes.

To assess the sex hormone-regulated LXA₄ expression in Kupffer cells, it is first necessary to check whether Kupffer cells express estrogen receptor, and what types of estrogen receptor they have. Estrogen receptor alpha is known to be mainly expressed in hepatocytes and receptor beta is expressed in hepatic stellate cells [Endocrinology. 1997; 138(3):863-70/ Genes Cells. 2015; 20(3):217-23/ Sci Rep. 2017 May 10; 7(1):1661/ J Gastroenterol Hepatol. 2018; 33(3):747-755]. The types of estrogen receptor expressed in Kupffer cells remains unclear although several papers have

suggested the possibility of expression of the estrogen receptors in Kupffer cells [J Immunol. 2009 Apr 1; 182(7): 4406./ Cell Immunol. 2003;222(1):27-34./ Surgery. 2006;140(2):141-8]. Thus, in order to investigate regulation of FPR2 agonist by estradiol, the types of estrogen receptors expressed in Kupffer cells first should be identified, and then hormone-controlled LXA₄ expression can be examined. However, the investigation is out of scope from our research, because we have proved that sex-specific expression of FPR2 is involved in hepa-protection from lipotoxicity and focused on estradiol-stimulated FPR2 in hepatocyte. In addition, hepatocytes are known to not produce LXA₄.

3. Some the data presented are not congruous, with significant differences - in some case - be evident at 36 hour time-point and other at different time points. This could be related to the different readouts under study. However, the Fpr2 expression does not seem different between males and females (Fig 3a and 3b).

: In qRT-PCR assay, we could not run all samples together because of the limited number of wells in qPCR machine. Hence, samples were analyzed separately for each time group, such as 6, 12, 36 weeks. Although the samples could not be compared among groups because of the separated analysis per treatment period, we used the same unit of Y axis to show the pattern of FPR2 expression among groups. As you commented below (comment #4), we reanalyzed the data using two-way ANOVA and also changed the unit of Y axis of each group according to data values generated from each group in the revised manuscript. In the corrected statistical analysis by two-way ANOVA, there was a significant difference between male and female mice at 6 weeks. The difference was observed in only chow-fed male and female mice in 12W group and disappeared in 36W groups.

4. The statistical assay applied is wrong. Many of the experimental designs have two variables, e.g. males and females, which change over time, e.g. 6, 12 and 36 weeks. Such designs would require a two-way ANOVA analysis to identify potential statistical differences

: As you pointed out, two-way ANOVA analysis is more proper to determine the statistical significance. Therefore, we reanalyzed the data using two-way ANOVA and corrected them in the revised manuscript. Also the explanation for statistical analysis method was revised.

Other ones:

1. Introduction (Page 1, Lines 46-50). What is the quantitative incidence of NALFD in man, premenopausal women, and post-menopausal women. Good to give an idea of the numbers.

: As you requested, we provided information of quantitative incidence rate of NAFLD in the revised manuscript. Although the trend of NAFLD incidence and prevalence in men and women according to age has been reported worldwide, specific numerical data are limited to a few countries in Asia, such as Korea, Japan, China. Hence, we provided the information obtained from these countries in the revised manuscript: **“In South Korea, the prevalence of NAFLD was higher in men than women, but it increases in post-menopausal women compared with premenopausal women (41.1% vs 25.8%/17%, respectively)⁷. In a cohort of Japanese subjects, the prevalence of NAFLD in men and postmenopausal women (24% and 15%, respectively) is higher than premenopausal women (6%)⁸. In South China, the prevalence of fatty liver disease, which includes 87% of NAFLD, is significantly lower in women than men under the age of 50 years**

(22.4% vs 7.1%, respectively), but women start to have higher prevalence rate compared to men over the age of 50 years (27.6% vs 20.6%)⁹.”

2. The word sex should be used instead of gender, which has more a societal connotation (e.g. Results, Page 1, Line 77). This is biology hence the focus is on sex differences.

: We appreciate your helpful comments for improving the quality of our manuscripts. We changed all “gender” with “sex” in the revised manuscript.

3. If there is a sex control on Fpr2 (which I find quite unlikely from the data presented) I would have expected differences to become evident at a earlier time-point and not 36 weeks (e.g. Results, Page 1, Line 95). By then there could be several intermediate mechanisms between sex hormones, liver inflammation, and outcome of NAFLD.

: QRT-PCR could not provide information of the expressional change of *Fpr2* during diet feeding because of the separate analysis per treatment period. However, protein analyses, western blot and IHC, allowed us to interpret the Fpr2 changes in all mice together. As you pointed out, difference of Fpr2 expression between male and female was most evident in at 6 weeks (Fig. 3a-c in the revised manuscript). Highest expression of Fpr2 in female mice at 6 weeks seems to exert the strongest protection effect and is expected to result in the most distinct liver response to injury compared with male mice. However, 6 and 36 week groups were fed CDAHFD for 6 and 36 weeks, respectively. 6-week-feeding brings to a mild injury, whereas 36-week-feeding, as a kind of chronic injury, results in a severe liver damage. Hence, it is hard to observe the apparent difference between male and female at mild injury. Given that NAFLD is progressive disease, NASH-like liver generated by chronic injury at 36 weeks after diet feeding is appropriate. During CDAHFD feeding, male mice develop NASH-like liver damage, whereas female mice seem to be resistant to the injury because hepatic FPR2 prevents female mice from developing NASH according to our findings (please see Fig. 3b-c in the revised manuscript. CDAHFD reduced Fpr2 expression, but the number of Fpr2-expressing cell is still significantly higher in female mice than in male mice). Although *in vivo* model reflects physiological response of the body, there are many factors impacting results. Hence, we conducted *in vitro* experiments to confine our findings in hepatocytes. Hepatocytes isolated from male mice rarely had Fpr2 expression, and were sensitive to PA injury (Fig. 7- 8). However, Fpr2 induction in these hepatocytes by estradiol improved their viability with function (Fig. 8). These findings suggest that FPR2 is involved in hepatocyte protection from lipotoxicity. As we described, this model was reported by Matsumoto’s group and we reproduced and confirmed the experimental model in our lab.

As you pointed out, there are a lot of mechanisms underlying NAFLD progression. And a lot of researchers have been studying for NAFLD. Even focusing on the present studies, there are several possible intermediate mechanisms explaining our findings. In the current research, we provide one mechanism among many possible mechanisms, and the mechanism is probably limited, however, it is impossible to fully explain everything in one research articles. Also, miR-181b is possibly interrupts Fpr2 expression in CDAHFD (Please see answer for comment #4), and it may be another factor, besides of estradiol, regulating FPR2 expression. It is the first step to find the Fpr2 expression in the livers of female mice and its expressional meaning in NAFLD. Further study will be conducted to find out the detailed mechanism and additional effect of FPR2 in other disease.

4. Which are the cell types expressing Fpr2 in the liver? How come the IHC expression seems to fade with ageing, e.g. female at 36 week (e.g. Results, Page 1, Line 143-144) when the only occasional statistical significance seems to be apparent at week 36? Unless the application of two-way ANOVA does not reveal different statistical differences. These two facts do not seem to be congruent.

Hepatocytes and Kupffer cells, a resistant macrophage in the liver, express Fpr2, shown by Fig. 3b, Fig. 7a-c and Supplementary Fig. 9. And hepatocytes seem to express Fpr2 at 36 weeks (Fig. 3b). Hence, to investigate the protective effect of FPR2 in hepatocytes from PA, we isolated these cells from mice.

As you pointed out, Fpr2 expression was faded as the female mice aged. Frankly, we do not know why Fpr2 staining is faded. The fading pattern was unique to Fpr2 protein, not other proteins which were examined in the present research, indicating that there was no technical problem. In many papers, stain intensity is measured and presented as protein expression level [J Hepatol. 2021 Apr 16;S0168-8278(21)00244-0./ Mol Syst Biol. 2020 Feb;16(2):e8985./ Hepatology. 2014 Oct;60(4):1264-77./ Hepatol Commun. 2019 Apr 25;3(7):925-942. Br J Pharmacol. 2018 May;175(9):1451-1470. Sci Rep. 2020 Feb 21;10(1):3201./ Front Med (Lausanne). 2020 Aug 11;7:450./ Cell Death Dis. 2017 Apr 13;8(4):e2748.].

Although Fpr2 expression was faded, Fpr2-expressing cells were still observed in chow-fed female group at 36W. The IHC results were supported statistically by the quantification data of Fpr2-positive hepatocytic cells (Fig. 3c-newly added data in the revised manuscript). Similar expressional change of Fpr2 was also observed in western blot analysis (Supplementary Fig. 4). Given that western blot measures protein amount in whole liver lysate, IHC showing morphology of cells allows us to center on Fpr2-expressing hepatocytic cells. Although quantification data of IHC could be slightly different from western blot data, but two different experiments generated similar results of Fpr2 expression. Hence, statistical significance of Fpr2 protein expression between chow-fed and CDAHFD-fed female mice is correct. However, we reanalyzed the RNA data using two-way ANOVA, statistical significance was not observed in RNA data of the 36W groups (Reviewer pointed out the statistical significance of *Fpr2* mRNA expression in 36W groups). Here is a problem; statistical significance of Fpr2 expression between chow- and CDAHFD-fed female groups was observed in protein data, not RNA data. In the previous version of manuscript, we just described Fpr2 expression without noticing statistical difference between protein and RNA data because we did not check statistical significance of Fpr2 protein. In the revised manuscript, we provided the possible explanation for this statistical difference in the discussion; **“Although *Fpr2* in the RNA level was not significantly different between the chow- and CDAHFD-fed female mice (Fig. 3a), the amount of Fpr2 protein was significantly decreased (Fig. 3b-c and Supplementary Fig. 4). The different expressional patterns of Fpr2 in RNA and protein levels indicate that CDAHFD might influence Fpr2 expression at post-transcriptional level. Pierdomenico et al.³⁸ have demonstrated that miR-181b directly binds to 3'-untranslated region of FPR2, and impacts FPR2 expression. In addition, it has showed that upregulated miR-181b lowers FPR2 expression by targeting FPR2, and reduced anti-inflammatory response caused by FPR2 suppression contributes to lung cystic fibrosis³⁹. Furthermore, miR-181b expression in serum was higher in the patient with NAFLD than healthy controls⁴⁰. Hepatic miR-181b was shown to be elevated in high fat diet-fed mice compared with normal diet-fed mice, and inhibition of miR-181b suppressed accumulation of triglycerides⁴⁰. In choline-deficient and amino acid defined (CDAA) diet**

model, miR-181b was upregulated in mice fed with CDAA⁴¹. Given that miR-181b targeting FPR2 is upregulated in the NAFLD-like experimental animal models and patients with NAFLD, and Fpr2 protein is lower in the CDAHFD-fed female than the chow-fed female mice, it is possible that CDAHFD decreases Fpr2 expression at post-transcriptional level by increasing miR-181b. However, Fpr2 amount was still elevated in females compared with males during CDAHFD feeding (Fig. 3b-c), indicating that estrogen-stimulated Fpr2 seems to at least partially compensate Fpr2 loss by CDAHFD, and prevents female mice from NAFLD progression. Further studies are required to verify the effect of miR-181b on FPR2 protein expression in NAFLD.”.

5. (e.g. Results, Page 2, Line 151). Is Fpr2 truly highly expressed in female livers? Fig. 3b would need some semi-quantitative analyses from several mice in order to have an idea of protein expression. IHC would need to identify the cells that express Fpr2.

: Because we could not see the expressional pattern of *Fpr2* according to treatment period by qRT-PCR assay, we wanted to check it using western blot. Hence, we pooled samples from 3 representative mice per each group (total 8 groups), and the pooled 8 samples were analyzed together. Also, it is impossible to load all samples together (3 mice X 8 groups=24 samples, and 1 marker) in one big gel because it provides only 20 wells. As we explained previously, individual protein expression was confirmed before pooling of protein lysates, and then equal amounts of protein lysates from representative mice were combined. As you requested, we provided these blot images showing Fpr2 in all mice in the revised manuscript (Supplementary Fig.4). In this case, quantitative analysis is impossible because male and female mice were separately examined (limited number of loading well), However, these blots clearly showed the Fpr2 expression in all mice during treatment.

To provide the quantification data of Fpr2 expression according to the treatment period, we used the Fpr2-stained liver section, because all liver sections from all mice were stained for Fpr2, and they were good experimental materials to analyze Fpr2 expression in all samples together without the limitation on the number of sample loadings. Hence we counted Fpr2-positive hepatocytic cells and graphed it with statistical analysis using two-way ANOVA in the revised manuscript (Fig. 3c): **“the number of Fpr2-positive hepatocytic cells was significantly higher in the female mice than the male mice during chow feeding (Fig. 3b-c). CDAHFD reduced the number of Fpr2-expressing cells in both male and female, but Fpr2 expression was significantly upregulated in female mice compared with male mice.”.**

In IHC for Fpr2, hepatocytes-looking cells were positive for Fpr2 in the healthy livers of WT female mice, and we provided the magnified images to show it. In addition, we isolated several liver-resident cells, such as hepatocyte, hepatic stellate cells, liver sinusoidal endothelial cells (LSEC) and Kupffer cells isolated from WT male and female mice, and examined what types of cells were positive for Fpr2. Western blot and/or qRT-PCR clearly revealed that hepatocytes and Kupffer cells expressed Fpr2 (Fig. 7a-c and Supplementary Fig. 9). Immunofluorescent staining (IF) for Fpr2 in these cells also supported these findings. In the revised manuscript, we added the double stained images of Albumin, a marker of hepatocyte, and Fpr2, as you requested. We confirmed that hepatocytes expressed Fpr2 (Fig. 7c). IF images were analyzed by confocal microscope. The method was described in the revised manuscript.

6. (e.g. Results, Page 4, Line 238). What is MT ERE?

: Sorry for forgetting to provide the full name. We provide it in the revised manuscript; mutant-type (MT) ERE. Full name of ERE was already explained.

7. Figures where time-courses and sexes are run over 36-week should highlight potential differences among the two sex groups, not compare the effect of time, as one presumes that the disease develops over time. In some cases there are too many comparisons, many of which not that relevant, e.g. Fig. 1b, Fig.5c and more.

: As you pointed out, comparison among the diet feeding period was intended to show the disease development, because we thought that it was necessary to show it. We totally agree with you. There are too many comparisons and it has made us to be nervous in analyzing data. We happily accepted your advice and compared the data among the sex groups. Therefore, we reanalyzed the statistical significance using two-way ANOVA and deleted confusing comparisons in the revised manuscript.

8. Similarly, many of the Figures present IHC data on representative images. It could be important to provide some quantitative data on images from different mice to understand the spread, and values, of the data.

: As you requested, the number of positive cells for active Caspase-3, Fpr2, F4/80 or CD68 was counted and graphed with statistical analysis using two-way ANOVA. The explanations for the data and method were added in the revised manuscript.

9. Figure 9a: it is strange that there is no difference in estradiol levels between the two diets? If so, how could one link pathological changes associated with the diets to Fpr2 control by sex hormones? Fig 9 seems to indicate that the changes in Fpr2 expression are not dependent from serum estradiol levels. Apology if I misunderstood this point.

: It is a really good point. Estradiol level was measured in serum, and Fpr2 expression was examined in liver tissue. Namely, their levels were assessed in the different source. Since CDAHFD directly damages the liver, Fpr2 expression in the liver is inevitably affected by CDAHFD. In the experiments conducted in female mice, serum estradiol levels were not significantly different between the chow and the CDAHFD group (Fig. 3d, Fig. 10a, and supplementary Fig. 16a) like that CDAHFD rarely affected serum estradiol level in male mice receiving exogenous E2 (Fig. 9a). As we mentioned above, miR-181 is possibly involved in reducing FPR2 expression. To figure out the direct regulation of FPR2 expression by estrogen, it should be examined in the intact liver. Fig. 3e presents a positive correlation between hepatic expression of *Fpr2* and serum estradiol levels in the chow-fed female mice. We inserted artificial estradiol pellets to examine whether estrogen “induced” Fpr2, and increased Fpr2 protected the liver from damage in males. Although hepatic Fpr2 expression had a tendency to be reduced slightly by CDAHFD, its level in the CDAHFD-fed E2 group was still significantly higher compared with the CDAHFD-treated P group. Without estradiol (such as male, placebo-supplemented male or OVX-female mice), Fpr2 level was remarkably low and resulted in the severe liver damage. However, Fpr2 induced by estradiol (such as female, E2-supplemented male, or OVX+E2-treated female mice) attenuated the liver injury. In line with the *in vivo* data, *in vitro* experiment showed that estrogen increased Fpr2 expression and protected hepatocytes from

lipotoxicity. Taken together, these results demonstrate that estrogen induces Fpr2 expression, which prevents the liver from NAFLD progression. These findings provide an explanation for sex-specific NAFLD development; As women aged, estrogen level decreased, and reduced FPR2 weaken the resistance to NAFLD development. Men with lower level of estrogen are sensitive to NAFLD development because of low level of FPR2.

Reviewer #3 (Remarks to the Author):

The MS entitled “Formyl peptide receptor 2 determines sex-specific differences in the progression of nonalcoholic fatty liver disease/steatohepatitis” and authorized by Chanbin Lee et al described that the presence of Formyl peptide receptor 2 regulated by estradiol in female mice protect from the development and progression of NAFLD.

Although the authors showed appealing results, the MS in the present form is quite descriptive. The authors need to identify the mechanism for which Formyl peptide receptor 2 expression in the liver boost the reduction of TG content.-

Major points:

1. First the authors should determine the levels of Formyl peptide receptor 2 in healthy human liver from male and female. RNA seq or arrays expression are already available in the literature. Also the expression of Formyl peptide receptor 2 should be analyzed in NAFLD patients with different sex.

: As you commented, we analyzed hepatic FPR2 expression in human using published microarray data to investigate the potential role of FPR2 in its clinical implications. We wanted to analyze the relationship of hepatic FPR2 with serum estrogen concentration, but the data which we want to have were not available. There are two separated data sets, adolescence (GEO access number: GSE66676) and adult group (GEO access number: GSE86932). However, we could not use the adult group, because age range of healthy women is too broad and 3 out of 5 healthy females were in their early 40's, and it is ambiguous age to determine reproductive status and estrogen reduction. On the other hand, adolescence group have an increasing level of estrogen in girls compared with boys. Hence, we used the data of adolescence group in the analysis. The explanation for the results were added in the revised manuscript; **“To determine if a similar correlation might be detected in human, we analyzed hepatic *FPR2* expression in human using published microarray data (GEO access number: GSE66676). The age of human included in the cohort ranges from 13 to 19 and represents the group having an increasing level of estrogen in girls compared with boys. Hepatic *FPR2* expression was significantly higher in healthy females than healthy males (Supplementary Fig. 5), while its level was dramatically reduced in female patients with NAFLD and similar with it in male patients. There was no difference of *FPR2* expression between healthy men and male patients with NAFLD.”**.

2. Also, it is necessary to determine if the promoter of the human gene Formyl peptide receptor 2 is also a target for estradiol or this is specific for the gene belong to the mice.

: As you requested, we examined whether estradiol regulated the human FPR2 expression by binding with FPR2 promoter region using luciferase reporter assay. We found that estradiol also bound to FPR2 gene of the human gene and added the data and explanation in the revised manuscript (Supplementary Fig. 10 in the revised manuscript); **“Estradiol also remarkably elevated luciferase activity in HepG2 cells transfected by pGL3 vectors containing ERE of promoter of human FPR2”**.

3. In figure 2 a deeper analysis should be performed. A more detailed profile of inflammatory response is required, as well as the evaluation of different hormones. In the MS, the authors should

evaluate the WAT and the BAT as well as the glucose metabolism. Additionally, food intake should be measured in these animals as well as the body weight during the timing 6, 12 and 36 weeks.

Moreover, DNL, beta oxidation and VLDL secretion should be evaluated in these mice.

: The sex-difference in the prevalence and incidence of NAFLD is strongly related to endogenous sex hormones. Testosterone is a primary sex hormone and play a critical role in the development of male reproductive tissue like testes and prostate in men. Several studies have reported that low testosterone is related with the risk for diabetes, obesity, and testosterone is considered as a factor contributing to the metabolic syndromes in men [J Androl. 2009; 30(1):10-22./ Nat Rev Endocrinol. 2013 Aug;9(8):479-93./ Diabetes Care. 2006 Mar;29(3):749; author reply 749-50/ Ther Adv Endocrinol Metab. 2010 Oct; 1(5): 207–223./ J Gastroenterol. 2006; 41(5):462-9.]. In women, high level of testosterone in serum is associated with higher risk of type 2 diabetes and hepatic steatosis, even that it enhances the risk of NASH and liver fibrosis in young pre-menopausal women, suggesting that testosterone may represent an early risk factor for NASH progression in young women, prior to their onset of more dominant, age-related metabolic risk factors [JAMA. 2006;295(11):1288-99/ Am J Gastroenterol. 2017; 112(5): 755–762./ Clin Gastroenterol Hepatol. 2021 Jun;19(6):1267-1274.e1.]. According to meta-analysis of human NAFLD data, men with NAFLD had significantly decrease testosterone level than healthy men did, whereas women with or without NAFLD at menopause had similar level of testosterone [Ann Hepatol. 2017;16(3):382-394./ Fertil Steril. 2018;109(4):728-734.e2 / Breast Cancer Res Treat. 2014;144(2):249-61.]. In the present studies, we suggest that increased incidence of NAFLD at women as aged is related with FPR2 because Fpr2 expression in the livers of female mice parallels with estradiol level, not testosterone, and we provide the evidence that FPR2 expression is regulated by estradiol and it protects the liver, especially hepatocytes from lipotoxicity.

Since the pathophysiology of NAFLD is associated with various organ including adipose tissue, muscle, intestine and liver, crosstalk between liver and the other tissues is important events, as you know. In the present study, we adopted the choline-deficient, L-amino acid-defined, high-fat diet (CDAHFD) to establish a human NAFLD-like animal model. The model was developed by Matsumoto's group [Int. J. Exp. Pathol. **94**, 93-103 (2013)/ Int. J. Exp. Pathol. **98**, 221-233 (2017)]. They showed that CDAHFD feeding for 12-60 weeks induced progression from steatosis to NASH with fibrosis and subsequently led to tumorigenesis in mice, suggesting that this model reflects human NAFLD/NASH better than other models. However, CDAHFD diet did not induced hypertrophy of visceral fat, adiposity, gain of body weight and peripheral insulin sensitivity because methionine to maintain visceral fat mass is minimally included in the diet [Int J Exp Pathol. 2013; 94(2): 93–103.]. Flores-Costa et al. reported that CDAHFD-fed male mice did not show the change of WAT mass compared with standard diet fed-mice [Proc Natl Acad Sci U S A. 2020; 117(45): 28263–28274]. In line with these reports, body weight decreased in CDAHFD-fed mice compared with chow-fed mice regardless of sex and FPR2 expression level in our animal model (body weight data was added in the revised manuscript. Supplementary Fig. 1b and Supplementary Fig. 7b). In addition, Börgeson et al. revealed that 3T3-L1 cells, mouse adipocyte cell line, did not express FPR2 and Hellmann et al. demonstrated that macrophages mainly expressed FPR2 and adipocytes accounted for minor part of FPR2 expression in adipose tissue [FASEB J. 2011; 25(7): 2399–2407]. Hence, most of the studies on FPR2 in adipose tissue center on the macrophages, not adipocytes. Based on our findings

and previous results shown by other groups, FPR2 in adipocytes seems to be minor in explaining the association of FPR2 with NAFLD.

Several researches have shown that CDAHFD-fed mice have lower food consumption whereas they have higher calorie intake values than normal diet-fed mice [PeerJ. 2019;7:e8115./ Nutrients. 2020;12(12):3886.] Based on these reports, it seems that CDAHFD-fed mice have low food consumption but high calorie intake than chow-fed mice. In addition, measurement of body weight and food intake are usually presented in the diet-induced obesity or insulin resistance mouse model focusing on the adipose tissue, rather than the liver [Nat Genet. 2010;42(12):1086-92./ Sci Rep. 2021;11(1):1323./ Nat Commun. 2020;11(1):2397./ Cell Death Dis. 2021;12(2):212./ Nature. 2018;555(7698):673-677.].

Influx of excessive free fatty acid (FFA) brings to oxidative stress and mitochondria dysfunction, resulting in massive hepatocytes death. And FFA level is known to be positively correlated with NAFLD severity [Gastroenterology. 2003 Aug;125(2):437-43./ J Hepatol. 2011 Jan;54(1):142-52./ Sci Rep. 2014 Jul 25;4:5832.]. Dying hepatocytes released several cytokines, such as PDGF, CTGF, TGF- β , and hedgehog, which promote inflammation and fibrosis in the liver. Thus, reducing hepatic FFA level is critical to treat NAFLD. Herein, we showed that FPR2 improved VLDL secretion, lowered lipid accumulation and ROS production in hepatocytes, increasing hepatocyte survival and functions (Fig. 8d-g and Supplementary Fig. 11). In addition, evidence that Fpr2 was involved in *Pemt* expression and VLDL secretion in vivo models was supported by TG levels in these animals (Fig. 1c, 4c, and Supplementary Fig. 12). Based on the effect of Fpr2 in hepatocytes, less inflammation is expected well because hepatocyte death brings to inflammatory response. To investigate inflammatory reactions, additional IHC for CD68, another marker of macrophages, was conducted. The number of F4/80 or CD68 was counted and plotted as the graph for quantitative comparison. CDAHFD induced significant increase of F4/80- or CD68-positive cells in WT mice, and male mice had great accumulation of these inflammatory cells compared with female mice (Fig. 2c-d and Supplementary Fig. 3). In KO mice, these cells were apparent in both sexes treated with CDAHFD, but there was no significant difference between male and female (Fig. 6c-d, and Supplementary Fig. 8). These results support that Fpr2 elimination further promotes the inflammatory response in the female liver.

We have studied the potential role of FPR2 in NAFLD because we first found the female-specific expression of Fpr2. Based that a positive correlation of *Fpr2* with estradiol in the healthy livers of female mice, we investigated the action of Fpr2 in NAFLD. As you questioned for the mechanism underlying the protective role of Fpr2 in the liver, we provided one possible mechanism explaining the effect of FPR2 in the fatty hepatocytes; PEMT regulation by FPR2. To present evidence that FPR2 is involved in lipid metabolism, specifically VLDL secretion through PEMT, we added the data for ROS, VLDL secretion, and lipid amount. These data support TG level in our animal model. The new data and explanation were added in the revised manuscript: Please see the answer for comment # 6.

4. In fig 3 the authors show changes in estradiol at 6 weeks, but no changes were detected at 12 and 36 weeks. However, Formyl peptide receptor 2 is express at different levels at 3, 12 and 36 hours with a positive tendency in comparison to the male mice. How the authors explain this lack of correlation between estradiol and Formyl peptide receptor

: In qRT-PCR assay, we could not run all samples together because of the limited number of wells. Hence, samples were analyzed separately for each time group, such as 6, 12, 36 weeks, and they could not be analyzed together. To provide the quantification data of Fpr2 expression according to the treatment period, we used the Fpr2-stained liver sections, because all liver sections from all mice were stained for Fpr2, and they were good experimental materials to analyze Fpr2 expression in all samples together without the limitation on the number of sample loadings. Hence, we counted Fpr2-positive hepatocytic cells and graphed it with statistical analysis using two-way ANOVA in the revised manuscript (Fig. 3c). As you pointed out, quantification data of Fpr2-expressing cells presented the significant difference between male and female mice during chow feeding, and estradiol level had a significant difference between two groups at 6 and 12 weeks. Considering mice age (7W+36W treatment=43-week-old), reduced level of estradiol in female mice and similar amount of estradiol among male and female mice at 36 weeks are correct. Estradiol level was measured in serum, and Fpr2 expression was examined in the liver tissue. Although serum estradiol level was not significantly different between male and female mice at 36 weeks, it was measured in serum, not the liver. In addition, estradiol level was presented as pg/mL, not relative level, but hepatic Fpr2 expression was shown as the fold increase compared with chow-male mice. Namely, Fpr2 level is expressed as a relative amount. Considering very low expression of Fpr2 in male mice, we excluded male mice and examined the correlation of estradiol with *Fpr2* in the chow-female groups.

5. Same approaches should be taken in fig 4 to determine inflammatory response, hormones, WAT and the BAT characterization, glucose metabolism, food intake as well as the body weight during the timing 6, 12 and 36 weeks. Also the mechanism underlying the regulation of lipid content in the KO mice is required to be analyzed. It has been previously reported the effect of Formyl peptide receptor 2 in the inflammatory response, but the TG content in the liver at 6 weeks is already different between male and female. The mechanisms underlying this effect should be analyzed and include in the present MS.

: Please see the answer for comment #3 above.

In regard of comment, “the TG content in the liver at 6 weeks is already different between male and female”

There was no significant difference between male and female mice at 6 weeks. However, CDAHFD-fed mice showed the increased TG level compared with the chow-fed mice. There was too many comparison for time-course and sex in the previous version, which brought to confusion. As other reviewer commented, we deleted some comparison which were irrelevant, and reanalyzed the data using two-way ANOVA in the revised manuscript.

6. In fig 8 in primary hepatocytes lipid context, mitochondrial ROS, inflammatory response and the mechanism underlying the lack of Formyl peptide receptor 2 should be evaluated.

: As you requested, we measured intracellular ROS levels using DCFH-DA. As expected, compared with PA-treated pHEPs from WT male mice without estradiol, estradiol-treated cells showed significant downregulation of ROS production, although these cells were exposed to PA. (Fig. 8f). In addition, compared with vehicle-treated pHEPs from KO male mice, PA-treated cells from these mice had higher amount of intracellular ROS, but estradiol treatment rarely reduced level of ROS in PA-exposed pHEPs from KO male mice. The data indicate that estradiol-induced Fpr2 protected

hepatocytes by reducing oxidative stress caused by PA. The data were included in the revised manuscript.

To explain how *Fpr2* protected hepatocytes from lipotoxicity, we investigated whether FPR2 was associated with lipid metabolism. Lipid context was assessed by Oil red O staining to determine if estrogen-induced *Fpr2* reduced lipid accumulation by affecting lipid metabolism (Supplementary Fig. 11). Choline is an essential nutrient in producing phosphatidylcholine (PC) required for VLDL synthesis. Deficiency of choline results in the excessive lipid storage in the liver by impaired lipid outflow from the liver. However, it has been reported that human and rodent female have a lower risk for hepatic steatosis than male, because of higher expression of hepatic PEMT in female. PEMT is a transferase enzyme, which makes PC from phosphatidylethanolamine in the liver. PEMT expression is directly regulated by estrogen, and lipid accumulation is less in female than male because of PEMT. Since we employed CDAHFD lacking choline and CDAHFD-fed mice had a defect in VLDL production and secretion (Supplementary Fig. 12), we investigated the association of PEMT and FPR2 in our model, and found that *Fpr2* influenced *Pemt* expression and improved VLDL secretion, reducing fat accumulation in hepatocytes and the liver. New data (Supplementary Fig. 11-12) and explanation were added in the revised manuscript: **“Choline deficiency impairs very-low density lipoprotein (VLDL) secretion, and leads to fat accumulation^{22,23}. However, female human and mice have a capacity for de novo biosynthesis of choline by phosphatidylethanolamine N-methyltransferase (PEMT), which makes them be less sensitive to choline deficiency^{24,25}. PEMT is known to be regulated by estradiol²⁶. Based on these findings, we examined whether FPR2 influenced VLDL secretion through PEMT, and reduced fat accumulation in hepatocytes, protecting them from PA injury. Estradiol-treated pHEPs from WT male mice containing higher *Fpr2* level showed significantly increased expressions of *Pemt* and VLDL secretion-related genes, apolipoprotein B (*ApoB*) and apolipoprotein C3 (*ApoC3*), compared with the vehicle-treated cells (Supplementary Fig. 11a). PA exposure downregulated *Pemt*, *ApoB*, and *ApoC3* in pHEP from WT males, but their expressions were significantly higher in the estradiol-treated pHEPs than the vehicle-treated cells. In pHEPs from KO mice, expressions of these genes were rarely induced by estradiol. In line with RNA data, Oil-red O staining showed that lipid droplets greatly accumulated in the PA-treated pHEPs from WT and KO male, whereas estradiol apparently lowered lipid droplets in PA-treated cells from WT mice, not from KO mice (Supplementary Fig. 11b). Given that estradiol hardly changed expressions of *Pemt* in pHEPs from KO mice, the data suggested that FPR2 might be involved in PEMT.**

Expressional changes of *Pemt* and VLDL secretion-related markers were also examined in the NAFLD-like animal models. As expected, the expressions of *Pemt*, *ApoB*, and *ApoC3* significantly increased in the chow-fed WT female mice compared with the chow-treated WT male (Supplementary Fig. 12a). CDAHFD reduced their levels in both sexes, but their expressions were significantly or tended to be elevated in the CDAHFD-fed females than CDAHFD-treated males. However, the RNA levels of these genes were similar in the KO female and KO male mice, regardless of diets (Supplementary Fig. 12b). KO female mice had a deficient *Fpr2*, not estradiol, and no expressional change of *Pemt* regulated by estradiol in these mice confirmed the *in vitro* data by indicating that *Fpr2* impacted *Pemt* expression.

Taken together, these results suggest that FPR2 is involved in VLDL secretion by affecting PEMT, and ameliorates liver damage.” in result section.

Influx of excessive free fatty acid (FFA) brings to oxidative stress and mitochondria dysfunction, resulting in massive hepatocytes death. And FFA level is known to be positively correlated with NAFLD severity [Gastroenterology. 2003 Aug;125(2):437-43./ J Hepatol. 2011 Jan;54(1):142-52./ Sci Rep. 2014 Jul 25;4:5832.]. Dying hepatocytes released several cytokines, such as PDGF, CTGF, TGF- β , and hedgehog, which promote inflammation and fibrosis in the liver. Thus, reducing hepatic FFA level is critical to treat NAFLD. Herein, we showed that Fpr2 improved VLDL secretion, lowered lipid accumulation and ROS production in hepatocytes, increasing hepatocyte survival and functions. In addition, evidence that FPR2 was involved in PEMT expression and VLDL secretion in vivo models was supported by TG levels in these animals. Based on the effect of Fpr2 in hepatocytes, less inflammation is expected well because hepatocyte death brings to inflammatory response. To check inflammatory response, additional IHC for CD68, another marker of macrophages, was conducted. The number of F4/80 or CD68 was counted and plotted as the graph for quantitative comparison. CDAHFD induced significant increase of F4/80- or CD68-positive cells in WT mice, and male mice had great accumulation of these inflammatory cells compared with female mice. In KO mice, these cells were apparent in both sexes treated with CDAHFD, and there was no significant difference between male and female. These results support that Fpr2 elimination further promotes the inflammatory response in the female liver. We described these findings and their relevance to the inflammatory response in discussion part: **“Choline is an essential nutrient in producing phosphatidylcholine (PC) required for VLDL synthesis⁴². Deficiency of choline results in the excessive lipid storage in the liver by impaired lipid outflow from the liver^{25,43}. However, it has been reported that human and rodent female have a lower risk for hepatic steatosis than male, because of higher expression of hepatic PEMT in female. PEMT is a transferase enzyme, which makes PC from phosphatidylethanolamine in the liver. PEMT expression is directly regulated by estrogen, and lipid accumulation is less in female than male because of PEMT²⁶. In the CDAHFD-fed WT mice, *Pemt* was significantly upregulated in WT female mice compared with WT male mice, and increased the expression of VLDL secretion markers, possibly reducing the levels of TG and hepatic fat (Supplementary Fig. 12a). However, sex difference of hepatic *Pemt* expression was disappeared in KO mice, and CDAHFD-fed KO female mice had similar degrees of lipid accumulation with CDAHFD-treated KO male mice did (Fig. 4a,c and Supplementary Fig. 12b). In addition, estradiol exposure induced *Pemt* expression and reduced accumulation of lipid droplets in pHEPs from WT male mice, not the cells from Fpr2-KO male mice (Supplementary Fig. 11). Based on these finding, it is possible that FPR2 is involved in PEMT regulation by estradiol, and improves VLDL secretion through PEMT, leading to the decreased fat accumulation in the liver. Influx of excessive free fatty acid (FFA) brings to oxidative stress and mitochondria dysfunction, resulting in massive hepatocytes death. And FFA level is known to be positively correlated with NAFLD severity^{16,44,45}. Dying hepatocytes released several cytokines, such as PDGF, CTGF, TGF- β , and hedgehog, which promote inflammation and fibrosis in the liver^{46,47}. Thus, reducing hepatic FFA level is critical to treat NAFLD. Therefore, it is possible that the promoting effect of FPR2 on VLDL secretion through PMET leads hepatocytes to be resistant to lipotoxicity, contributing to the hepatocyte survival, subsequently reduction of inflammation and fibrosis.**

However, further studies are required to investigate the detailed interaction among FPR2, PEMT and estrogen in modulating lipid metabolism in the liver.”.

7. Finally, in Fig 10, it will be relevant to identify if estradiol treatment will increase the levels of Formyl peptide receptor 2 in OVX mice.

: As you requested, we conducted the additional experiments of E2 recovery in female mice with OVX. Before feeding diet, WT female underwent ovariectomy and took a recovery period of one week (Supplementary Fig. 15). And then these mice received E2 or placebo pellet in the mid-ventral subcutaneous region. One week after E2 supplementation, these female mice were fed either chow or CDAHFD diet for 12 weeks. The explanation for the experimental method was added in the revised manuscript; **“In addition, E2 pellets were given to female mice receiving OVX to double check the action of estradiol-mediated Fpr2 in the liver. Briefly, female mice were treated with estradiol (OVX-E2) (n=10) or placebo pellet (OVX-P) (n=10) at one week after they (5-week-old) underwent ovariectomy. Post one week after supplementation, these female mice were divided into randomly four experimental groups and fed Chow or CDAHFD for 12 weeks: Chow-OVX-P (n=4), CDAHFD-OVX-P (n=6), Chow-OVX-E2 (n=4), and CDAHFD-OVX-E2 (n=6). At the end of each time point, mice were sacrificed to collect blood and liver samples”.**

The description for the results was added in the revised manuscript; **“To double check the action of estrogen-regulated Fpr2 expression in NAFLD, ovariectomized WT female mice were supplemented estradiol (OVX-E2) or placebo pellets (OVX-P), then were fed chow or CDAHFD for 12 weeks (Supplementary Fig. 15a). Reduced levels of serum estradiol and hepatic Fpr2 expression significantly increased in the OVX-E2 compared with the OVX-P groups (Supplementary Fig. 15f and 16a-c). Compared with the CDAHFD-fed OVX-P group, the CDAHFD-treated OVX-E2 group had the reduced liver damage and accumulation of Caspase 3-positive cells (Supplementary Fig. 15b-d and 16d). Estradiol supplementation also mitigated the enhanced hepatic fibrosis and inflammation in CDAHFD-given female mice with OVX (Supplementary Fig. 15e-g and 16d-f). In the chow groups, hepatic injury by estradiol supplement were absent (Supplementary Fig. 15-16).”**

Reviewers' Comments:

Reviewer #2:

Remarks to the Author:

The Authors have gone some length to try and address the several questions received, including the points I raised. I appreciate and acknowledge the extra experiments performed.

I sense that the question on the FPR2 agonist has been missed, or not properly explained, by me. The fact that a receptor is more or less expressed (ideally at protein level rather mRNA level) can be ineffectual unless this receptor encounters its agonists. So the fact that in vitro addition of lipoxin A4 modulates FPR2 expression is of some interest, but does not address the question. In other words, what can one measure in liver homogenates with respect to FPR2 agonists: LXA4, RvD1, AnxA1 and/or SAA? All these mediators can be measured by ELISA. If, for example, AnxA1 is detected and possibly changed between the two conditions (diet, or oestrogen) then immunohistochemistry can be used to pin-point the expression pattern of the agonist. The modulation of FPR2 expression by oestrogen remains important.

If the paper is accepted, I would suggest the in-vitro data with LXA4 are presented. However, I have indicated above what one could go about to understand the mechanism evolving around FPR2 to provide liver protection.

Reviewer #3:

Remarks to the Author:

the authors addressed all the concerns raised for the reviewer.

In future, the authors should further evaluate the role of PEMT in their animal model

Reviewer #4:

Remarks to the Author:

The authors have done a great job in revising their manuscript and including several new data (E2 supplementation in OVX, human expression data, etc.). This reviewer is satisfied with the revised manuscript in response to the original comments. However, I have few minor suggestions.

- Abstract, line 25: replace women with premenopausal women.

- Line 355: replace ascent with absent.

- Immunoblots of FPR2 in the right panel of supplemental figure 6b were not convincing. This reviewer could see faint bands of FPR2 in the KO females unless they are non-specific bands. Maybe adding the IHC data of the KO mice (shown in the rebuttal) in the supplemental figure 6 could alleviate this.

Reviewer #2 (Remarks to the Author):

The Authors have gone some length to try and address the several questions received, including the points I raised. I appreciate and acknowledge the extra experiments performed.

I sense that the question on the FPR2 agonist has been missed, or not properly explained, by me. The fact that a receptor is more or less expressed (ideally at protein level rather mRNA level) can be ineffectual unless this receptor encounters its agonists. So the fact that in vitro addition of lipoxin A4 modulates FPR2 expression is of some interest, but does not address the question. In other words, what can one measure in liver homogenates with respect to FPR2 agonists: LXA₄, RvD1, AnxA1 and/or SAA? All these mediators can be measured by ELISA. If, for example, AnxA1 is detected and possibly changed between the two conditions (diet, or oestrogen) then immunohistochemistry can be used to pin-point the expression pattern of the agonist. The modulation of FPR2 expression by oestrogen remains important.

Thank you for your kind explanation. As you requested, we assessed the hepatic level of LXA₄ using ELISA. As we replied to the previous comments, LXA₄ expressed by Kupffer cells has been shown to reduce inflammation in the liver. Hence, we chose and measured LXA₄ amount, and found that LXA₄ expression did not have a significant difference between male and female mice. However, its level was significantly alleviated in the damaged livers of both male and female mice compared with the healthy livers of the chow-fed mice. Lower expression of LXA₄ in the CDAHFD groups than the chow groups was also observed in Fpr2 KO mice. In the experiments regulating estrogen level, such as supplementation of exogenous estrogen, removal ovary, and supplementation of exogenous estrogen in ovariectomized mice, LXA₄ was downregulated in the CDAHFD groups compared with the chow groups. The expressional pattern of LXA₄ was related with the injury, not a sex. These findings indicate that LXA₄ is expressed in the livers of both male and female, and FPR2, a receptor for LXA₄, is highly present in the livers of female mice only, suggesting that female-specific expression of FPR2, not male mice, effectively reduce CDAHFD damage by inducing PEMT expression. PEMT data were provided in the previous revised manuscript. The data and explanation were added in the revised manuscript (Supplementary Fig. 17): “In addition, we measured the level of LXA₄, one of FPR2 ligands, in our experimental animal models. LXA₄ has been shown to have anti-inflammatory effects in the various disease models including acute liver failure and obesity^{13,33,34}. Hepatic LXA₄ amount rarely had a significant difference between male and female mice (Supplementary Fig. 17a). However, LXA₄ was downregulated significantly in CDAHFD-fed male and female mice compared with chow-fed mice. Decrease of LXA₄ expression in the CDAHFD groups compared to the chow groups was also found in the experimental models in which Fpr2 was knockout or the estrogen level was altered artificially (Supplementary Fig. 17b-e). The data indicated that both male and female mice had LXA₄, but female mice with higher level of Fpr2, LXA₄ receptor, could effectively reduce the liver damage”

We could not provide immunohistochemistry data of LXA₄ in our model because anti-LXA₄ is not commercially available.

If the paper is accepted, I would suggest the in-vitro data with LXA₄ are presented. However, I have indicated above what one could go about to understand the mechanism evolving around FPR2 to provide liver protection.

In the previous rebuttal, we presented that the exogenous LXA₄ induced FPR2 expression in hepatocytes from male mice, and induced FPR2 protected these cells from lipotoxicity. Although endogenous hepatic level of LXA₄ did not show the significant difference between male and female, its level tended to be higher in female mice than male mice at 6 weeks ($p=0.064$). However, the data of LXA₄ from other experimental groups, such as groups of 12 and 36 weeks, KO groups, and

estrogen altered groups, did not show any significant difference in sex or estrogen level. These results indicate that female mice expressing Fpr2 are resistant to NAFLD progression by effectively responding to LXA₄, whereas male mice not having Fpr2 hardly respond to LXA₄ and are sensitive to NAFLD development, supporting the importance of FPR2, “No matter how many ligands there are, they are meaningless if there are no receptor responding to them”

In line with other findings showing the anti-inflammatory role of exogenous LXA₄ in the livers of male mice, we found that exogenous LXA₄ artificially induced Fpr2 in hepatocytes from male mice and protected these cells from damage. However, we measured the endogenous level of LXA₄ and endogenous amount of LXA₄ seemed to be not enough to protect the liver from injury. To add in vitro data which reviewer suggest to include in the manuscript, we need to investigate the effect of exogenous LXA₄ in all experiments (all *in vivo* models) which we have conducted. It will be another huge experiments for LXA₄ functions, and generate the massive data. It appears to be out of scope. Considering the current data of endogenous LXA₄ in the liver (in vivo) and of exogenous treatment of LXA₄ (in vitro) and the irrelevance of LXA₄ with estrogen (or sex), it is difficult to link them, even bring to confusion. Hence, exogenous in vitro data and explanation were deleted and endogenous LXA₄ data were added in the revised manuscript.

We hope you generously understand our intention.

Reviewer #3 (Remarks to the Author):

the authors addressed all the concerns raised for the reviewer.

In future, the authors should further evaluate the role of PEMT in their animal model

Thank you for your comments. We will investigate the role of PEMT in our animal models in future.

Reviewer #4 (Remarks to the Author):

The authors have done a great job in revising their manuscript and including several new data (E2 supplementation in OVX, human expression data, etc.). This reviewer is satisfied with the revised manuscript in response to the original comments. However, I have few minor suggestions.

- Abstract, line 25: replace women with premenopausal women.

Yes, we replaced it.

- Line 355: replace ascent with absent.

Sorry for our mistake. We corrected it.

- Immunoblots of FPR2 in the right panel of supplemental figure 6b were not convincing. This reviewer could see faint bands of FPR2 in the KO females unless they are non-specific bands. Maybe adding the IHC data of the KO mice (shown in the rebuttal) in the supplemental figure 6 could alleviate this.

Supplementary Fig. 6 aimed to show the absence of Fpr2 expression in the livers of KO mice fed either chow or CDAHFD using qRT-PCR and western blot. Because we presented the expression of Fpr2 protein in individual WT mouse in Supplementary Fig. 4, we replaced the blot images showing Fpr2 in WT and KO mice (Supplementary Fig. 6b, right panel, in the previous version) with IHC data of Fpr2 in KO mice in the revised manuscript (Supplementary Fig.6c) instead of providing IHC data in the rebuttal, based on your comment. In addition, we deleted qRT-PCR data in bottom panel of Supplementary Fig. 6a because the data were duplicated with the data in top panel of Supplementary Fig. 6a. The difference between them is just different arrangement of same data. Thanks.

Reviewers' Comments:

Reviewer #2:

Remarks to the Author:

None.

Reviewer #4:

Remarks to the Author:

I am very satisfied by the answers to my comments. I would recommend the editor and the journal to 'Accept' the manuscript.

REVIEWERS' COMMENTS

Reviewer #2 (Remarks to the Author):

None.

: Thank you for your time in reviewing our manuscript.

Reviewer #4 (Remarks to the Author):

I am very satisfied by the answers to my comments. I would recommend the editor and the journal to 'Accept' the manuscript.

: Thank you for your time.